



# HCLIM38: A flexible regional climate model applicable for different climate zones from coarse to convection permitting scales

Danijel Belušić[1], Hylke de Vries[2], Andreas Dobler[3], Oskar Landgren[3], Petter Lind[1], David Lindstedt[1], Rasmus A. Pedersen[4], Juan Carlos Sánchez-Perrino[5], Erika Toivonen[6], Bert van Ulft[2], Fuxing Wang[1], Ulf Andrae[1], Yurii Batrak[3], Erik Kjellström[1], Geert Lenderink[2], Grigory Nikulin[1], Joni-Pekka Pietikäinen[6], Ernesto Rodríguez-Camino[5], Patrick Samuelsson[1], Erik van Meijgaard[2], and Minchao Wu[1]

[1]Swedish Meteorological and Hydrological Institute (SMHI), Norrköping, Sweden
[2]Royal Netherlands Meteorological Institute (KNMI), De Bilt, the Netherlands
[3]Norwegian Meteorological Institute (MET Norway), Oslo, Norway
[4]Danish Meteorological Institute (DMI), Copenhagen, Denmark
[5]Agencia Estatal de Meteorología (AEMET), Madrid, Spain
[6]Finnish Meteorological Institute (FMI), Helsinki, Finland

**Correspondence:** Danijel Belušić (danijel.belusic@smhi.se)

**Abstract.** This paper presents a new version of HCLIM, a regional climate modelling system based on the ALADIN-HIRLAM numerical weather prediction (NWP) system. HCLIM uses atmospheric physics packages from three NWP model configurations, HARMONIE-AROME, ALARO and ALADIN, which are designed for use at different horizontal resolutions. The main focus of HCLIM is convection permitting climate modelling, i.e. developing the climate version of HARMONIE-AROME. In HCLIM, the ALADIN and ALARO physics packages are used for coarser resolutions where convection needs to be parameterized. Here we describe the structure, development and performance of the current recommended HCLIM version, cycle 38.

HCLIM38 is a new system for regional climate modelling and it is being used in a number of national and international projects over different domains and climates, ranging from equatorial to polar regions. Our initial evaluation indicates that HCLIM38 is applicable in different conditions and provides satisfactory results without additional region-specific tuning.

HCLIM is developed by a consortium of national meteorological institutes in close collaboration with the ALADIN-HIRLAM NWP model development. While the current HCLIM cycle has considerable differences in model setup compared to the NWP version (primarily in the description of the surface), it is planned for the next cycle release that the two versions will use a very similar setup. This will ensure a feasible and timely climate model development and updates in the future and provide evaluation of long-term model biases to both NWP and climate model developers.

## 1 Introduction

Regional climate models (RCM) are currently used at a variety of spatial scales and model grid resolutions. Since the main motivation of using RCMs is to add value compared to global climate models (GCM) by downscaling over a limited area





domain, the resolution of RCMs is several times higher than that of GCMs (e.g., Rummukainen, 2010). The added value comes from the improved resolution of both regional physiography and atmospheric processes. The GCM resolution is typically O(100 km), while RCMs use resolutions of O(10 km) (e.g., Taylor et al., 2012; Jacob et al., 2014). At the same time, there is both evidence that O(10 km) is too coarse for resolving some important physical processes and a growing demand for even higher

resolution climate information by end users. Building on development and usage of numerical weather prediction (NWP) and research models at resolutions of O(1 km), and with the increase in available computational resources, climate simulations are increasingly being performed at those very high resolutions (e.g., Ban et al., 2014; Prein et al., 2015; Kendon et al., 2017; Coppola et al., 2018; Lenderink et al., 2019). An important conceptual change occurs between the resolutions of O(10 km) and O(1 km), where the parameterizations of deep convection that are used at O(10 km) or coarser resolutions are typically not

employed at O(1 km). The reason for the latter lies in the fact that the most important deep convection processes occur at scales of O(1 km) or larger and therefore the models at those resolutions - having also non-hydrostatic dynamics - should be able to resolve them. Consequently, climate models at resolutions of O(1km) are often referred to as convection permitting regional climate models (CPRCM). This still leaves a subset of smaller scale convection features that need to be parameterized, which is usually done using a shallow convection parameterization. Despite the extensive experience with such modelling systems

for NWP or research purposes, there are additional conceptual and computational challenges when climate simulations are considered. Here we present the new version (cycle 38) of the HARMONIE-Climate (HCLIM hereafter; Lindstedt et al. (2015); Lind et al. (2016)) modelling system, aimed at regional climate simulations on convection-permitting scales. Because the step from GCM to CPRCM resolution is still too large for direct nesting (e.g., Matte et al., 2017), there is a need for an intermediate model grid between GCM and CPRCM. HCLIM has options for this as described below.

HCLIM cycle 38 (HCLIM38) has been developed and maintained by a subset of national meteorological institutes from the HIRLAM consortium: AEMET (Spain), DMI (Denmark), FMI (Finland), KNMI (the Netherlands), MET Norway, and SMHI (Sweden). HCLIM38 is the new recommended version with considerable changes and improvements compared to the older versions. It is currently being used in a number of projects, some of which are large collaborative activities (e.g. H2020 European Climate Prediction system (EUCP; https://www.eucp-project.eu/), CORDEX Flagship Pilot Study (FPS) on

convection (Coppola et al., 2018), ELVIC CORDEX FPS on climate extremes in the Lake Victoria Basin (http://www.cordex. org/endorsed-flagship-pilot-studies/)).

The purpose of this paper is to describe the HCLIM38 modelling system, the available model configurations and physics packages, and to provide initial insight into their performance. HCLIM38 is closely related to the documented ALADIN-HIRLAM NWP modelling system and therefore this paper focuses on the features that distinguish the climate modelling

system from its NWP counterpart. Extensive evaluations of the different HCLIM38 model configurations and for different regions will be presented in separate studies.





## 2 Modelling system description

### 2.1 HCLIM structure and terminology

HCLIM is a regional climate modelling system based on the ALADIN-HIRLAM (see Table 1) NWP system (Lindstedt et al., 2015; Bengtsson et al., 2017; Termonia et al., 2018) . HARMONIE is a combination of the scripting system used for operational

NWP applications within HIRLAM countries and the specific configuration of AROME described in Bengtsson et al. (2017). The only supported atmospheric physics package in the HARMONIE system is AROME, hence the model configuration is frequently referred to as HARMONIE-AROME. This also distinguishes it from the AROME-France configuration used by Météo-France (Seity et al., 2011). HARMONIE-AROME is the basis used for HCLIM development. AROME is designed for convection permitting scales and is used with non-hydrostatic dynamics.

Two other atmospheric physics packages are available within the ALADIN-HIRLAM system (and hence HCLIM), namely ALADIN and ALARO (see Table 1). These are typically used with hydrostatic dynamics in HCLIM applications. In total, HCLIM includes three different atmospheric physics packages, making it necessary to specify which package is being used for a given application. Together with specifying the cycle used (cycle 38 in this paper), this results in the final terminology that is used in projects: HCLIM38-AROME, HCLIM38-ALADIN, or HCLIM38-ALARO.

As seen above, HCLIM versions are called cycles to remain consistent with the NWP model configurations. The HCLIM community strives to keep up with the NWP cycle releases, but due to the different time scales and applications of climate simulations, smaller development community and other various reasons there can be skipped cycles so gaps in cycles are expected in HCLIM terminology. For example, the HCLIM release before cycle 38 that was described in detail was cycle 36 (Lindstedt et al., 2015), and the next target cycle is 43.

### 2.2 HCLIM model configurations

Unlike the majority of limited area models, a common characteristic of all model configurations in the HCLIM system is that they use a bi-spectral representation for most of prognostic variables, while the spatial and temporal discretization uses a semi-implicit semi-Lagrangian scheme. The details of the dynamics are described in Bengtsson et al. (2017) and Termonia et al. (2018). The three different model configurations available in HCLIM are designed for different spatial resolutions (Fig.

1). A comprehensive description of the HARMONIE-AROME physics and NWP setup is presented in Bengtsson et al. (2017). Consequently, we here report only the main features and the differences between HCLIM38-AROME and the similar NWP system. The other two physics packages, ALADIN and ALARO, are detailed in Termonia et al. (2018), and here we describe the differences in HCLIM38. It is important to note that Bengtsson et al. (2017) and Termonia et al. (2018) describe the canonical model configurations for HARMONIE-AROME, and ALADIN and ALARO, respectively. Canonical model configurations

are a subset of all possible configurations that are thoroughly validated for use in a certain NWP context. The configurations presented here differ from the canonical model configurations. One important difference is the surface model, which is shared between all the three HCLIM model configurations. It is based on the surface model SURFEX (Surface Externalisée), with



different options activated for climate applications compared to the NWP setup as described below.

### 2.2.1 SURFEX

SURFEX is an externalized surface modelling system that parameterizes all components of the surface (Masson et al., 2013). It

uses a tiling approach to represent subgrid surface heterogeneity, with surface split into four tiles: continental natural surfaces, sea, inland water and town. The SURFEX tiling approach assumes that the interaction of each surface tile with the overlying atmosphere is completely independent of the other tiles present in the grid box. Continental natural surfaces are further divided into subtiles or patches, accounting for different vegetation characteristics within a grid box. The number of patches can be chosen between 1 and 12, but for simplicity it is typically set to 2 in HCLIM38, representing open land and forest.

The NWP setup of HARMONIE-AROME (Bengtsson et al., 2017) uses simplified surface processes like the force-restore approach in the soil (Boone et al., 1999) and the composite snow scheme by Douville et al. (1995). Such simplified physics is appropriate for short time scales in combination with surface data assimilation, but for climate time scales more sophisticated surface physics is required that can represent e.g. long soil memory and proper snow characteristics. The default SURFEX version in HCLIM38 was 7.2 (Masson et al., 2013). However, the state of the more sophisticated surface physics options in

v7.2 was not considered as adequate for HCLIM purposes. Therefore a stepwise upgrade of SURFEX was performed. First v7.2 was replaced by v7.3 and then the land-surface physics scheme in v7.3 (ISBA) was replaced by the corresponding scheme from v8.1. The list of SURFEX parameterizations and the related references used in HCLIM38 is given in Table 2. These include e.g. the ISBA multi-layer soil diffusion option (ISBA-DIF) with soil organic carbon taken into account and a 12-layer Explicit Snow scheme (ISBA-ES). For a proper simulation of temperature in deep and large lakes the inland water (rivers and

lakes including any ice cover) is simulated by Flake. In HCLIM38, FLake uses the lake depth obtained from the Global Lake Data Base (GLDB; Kourzeneva et al. (2012)) and is initialized from a climatology of lake profiles which vary over time and space. HCLIM38 implements the same SICE model as in the NWP setup to simulate the sea ice temperature. The difference in HCLIM38 is that the sea surface temperature and sea ice concentration are updated together with the lateral boundaries to capture their long term variation. The urban tile is parameterized by the Town Energy Balance model (TEB; Masson (2000)),

which is used only at convection permitting resolutions.

The land cover and soil properties are obtained from the ECOCLIMAP version 2.2 database at 1 km resolution (Faroux et al., 2013) and the Food and Agriculture Organization database (FAO, 2006), respectively. SURFEX is fully coupled to atmospheric model, receiving atmospheric forcing over each patch and tile and returning grid averaged values (heat and momentum fluxes, etc.).

### 2.2.2 ALADIN

ALADIN is the default choice in HCLIM38 for simulations with grid spacing close to or larger than 10 km. It is the limited-area version of the global model ARPEGE, from which it inherits all dynamics and physics options. The version of ALADIN available in HCLIM38 corresponds to the one used in NWP, with the dynamics and physics options listed in Table 3 and





described in Termonia et al. (2018). The difference in HCLIM38 is only in the surface parameterizations, which are more suitable for climate simulations as described above. HCLIM38-ALADIN is predominantly used as a hydrostatic model with a convection parameterization, so it is not envisaged for use at grid spacings much smaller than 10 km. Therefore, there is a gap, usually termed "the grey zone" for convection, between the grid spacing of 10 km and the convection permitting scales with

grid spacing smaller than 4 km. HCLIM38 simulations with grid spacing within the grey zone should be avoided if possible.

### 2.2.3 ALARO

ALARO has been designed to operate also in the convection grey zone (Fig. 1; Termonia et al. (2018)). Unlike traditional moist convection parameterizations, the Modular Multiscale Microphysics and Transport scheme (3MT; Gerard et al. (2009)) addresses the fact that the size of convective cells becomes significant compared to the model grid spacing as the resolution

increases. This allows for great flexibility in applying the model, which is highly desirable in climate applications. Therefore, ALARO was the default choice in HCLIM36 (Lindstedt et al., 2015). However, the coupling of ALARO with SURFEX as used in HCLIM38 resulted in too weak evaporation from oceans in low latitudes, causing considerable underestimation of atmospheric moisture content and consequently lack of precipitation in long climate simulations. Similar issues have not been observed at high latitudes (Toivonen et al., 2019) implying that ALARO can be used there. However, it is not the default option

in HCLIM38 because the modelling system should be applicable at all latitudes. The detailed ALARO description in Termonia et al. (2018) refers to a newer version called ALARO1. HCLIM38 uses an older version ALARO0 with some updates from a development version of ALARO1, and as such does not correspond to any canonical model configuration. Specifically, in HCLIM38-ALARO the radiation scheme ACRANEB from ALARO0 was replaced by an early version of ACRANEB2 used in ALARO1. Since it is based on ALARO0, HCLIM38-ALARO still uses the pseudo-prognostic TKE (pTKE) turbulence scheme

(Geleyn et al., 2006), which has been replaced by TOUCANS (Ďurán et al., 2014) in ALARO1. At the same time, HCLIM38-ALARO is coupled to SURFEX making this configuration different from the NWP model configuration. The full list of used parameterizations and references is given in Table 3.

### 2.2.4 AROME

HCLIM38-AROME is a nonhydrostatic CPRCM based on the HARMONIE-AROME NWP model configuration (Table 3), but

with different surface model options that are more suitable for climate applications as described above. It is the recommended option in HCLIM38 for simulations at convection permitting scales. AROME is the main focus of HCLIM development due to the recognized need of the participating institutes for a CPRCM. As such, it is becoming the backbone of convection permitting regional climate projections for a number of European countries.

There is no deep convection parameterization in AROME, so it can only be used at convection permitting resolutions with

grid spacing smaller than 4 km. This implies that for downscaling of low-resolution GCMs, it is not possible to use the same physics package at intermediate scales of O(10 km) and convection permitting scales of O(1km). As described above, the intermediate scales in the current setup are typically simulated by HCLIM38-ALADIN. This is the preferred option because the two physics packages share the same surface model. In this case the soil state in HCLIM38-AROME is initialized in



a consistent way, the only difference being the resolution change. This helps in decreasing the soil spin-up time. However, HCLIM38-AROME can be used also with different RCMs at the lateral boundaries.

HCLIM38-AROME uses a shallow convection parameterization based on the eddy diffusivity mass-flux framework (EDMFm; de Rooy and Siebesma (2008); Bengtsson et al. (2017)). The default choice for the turbulence parameterization is the scheme called HARMONIE with RACMO Turbulence (HARATU; Lenderink and Holtslag (2004); Bengtsson et al. (2017)), even though the CBR scheme (Cuxart et al., 2000) with the diagnostic mixing length from Bougeault and Lacarrere (1989) is available as well. HARATU and CBR mostly differ in the formulation of length scales and values of constants (Bengtsson et al., 2017). A one-moment microphysics scheme ICE3 (Pinty and Jabouille, 1998; Lascaux et al., 2006) is used, with additional modifications for cold conditions called OCND2 (Müller et al., 2017). The cloud fraction is determined using a statistical scheme (Bechtold et al., 1995). Similarly to ALADIN, the radiation scheme is a simplified version of the scheme used at ECMWF, described in Mascart and Bougeault (2011). The six spectral band short-wave radiation (SW6) is based on Fouquart and Bonnel (1980). Rapid Radiative Transfer Model (RRTM) with 16 spectral bands is used for long wave radiation (Mlawer et al., 1997; Iacono et al., 2008). Monthly aerosol climatologies are provided by Tegen et al. (1997).

### 2.3 Differences from Météo-France/CNRM climate models

Since the names of the physics packages are the same for HCLIM and Météo-France/CNRM climate models (CNRM-ALADIN and CNRM-AROME), we briefly describe some of the main differences between the latest versions of HCLIM and CNRM models (see Table 3 for the parameterization schemes used in HCLIM38).

CNRM's latest version of ALADIN-Climate, version 6.3 (CNRM-ALADIN63; Daniel et al. (2019)), differs from HCLIM38-ALADIN in certain physical parameterizations. Firstly, the parameterizations for convection and clouds are different: CNRM-ALADIN63 includes a PCMT scheme (Piriou et al., 2007; Guérémy, 2011) for dry, shallow and deep convection, and a PDF-based cloud scheme by Ricard and Royer (1993). Orographic wave drag is parameterized by Déqué et al. (1994) along with Catry et al. (2008) that is used in HCLIM38-ALADIN. A climate version of SURFEX 8 (Decharme et al., 2019) is used as the land surface modelling platform. It is also worth noting that when CNRM-ALADIN is coupled with e.g. aerosols (Drugé et al., 2019), oceans and rivers, the model is called CNRM-RCSM (regional climate system model) even if the schemes for the atmosphere are the same as described above.

The CNRM-AROME41t1 model is currently used for climate research (Coppola et al., 2018). This latest version 41t1 uses the cycle 41 of the physics package AROME that is described in Termonia et al. (2018) for the NWP system. The differences from HCLIM38-AROME are as follows. The ICE3 scheme is used for microphysics without the OCND2 modification, and the PMMC09 scheme is used for shallow convection (Pergaud et al., 2009). Parameterizations by Pergaud et al. (2009) are also used for clouds in addition to Bechtold et al. (1995) that is used in HCLIM38-AROME. The surface scheme is the version 7.3 of SURFEX (Masson et al., 2013). In addition, CNRM-AROME41t1 includes the COMAD correction for overestimated precipitation and unrealistic divergent winds in the vicinity of convective clouds (Malardel and Ricard, 2015).





## 2.4 Differences from HCLIM36

The version of the latest operational HCLIM was cycle 36, described thoroughly by Lindstedt et al. (2015). In the new version, cycle 38, the model system has undergone considerable changes in the parameterizations, physical packages used and target resolutions. The main differences are briefly described here for easier distinction between the different cycles.

In HCLIM38, ALADIN is the default model configuration that should be used for coarser resolution applications and as the intermediate model when the target horizontal grid size is on the kilometer scale or below. In the previous cycles the default configuration was ALARO. The other main difference is the update of the surface modelling platform going from SURFEX v5 to a blend of SURFEX v7 and v8, in order to improve and enable more accurate land related processes. The main improvements in SURFEX processes in cycle 38 are summarized in the following. Similarly to the differences from the
current NWP configuration (see Section 2.2.1), a 14-layer soil diffusion scheme together with a 12-layer explicit snow scheme are activated with soil organic carbon taken into account. The inland water is simulated by the lake model FLake. The sea ice model SICE is employed for oceans, and this model updates sea ice concentrations and sea surface temperatures with the same frequency as the lateral boundaries. The combined effects from the differences in model configurations lead to considerable improvement in the overall performance. For instance the near surface temperature is much closer to observations (Fig. S1a),
mainly dependent on the utilization of the diffusion soil scheme together with the sea ice model. Also, the general precipitation pattern is largely improved, especially during the summer, when the old configuration suffered from a very dry bias in eastern Europe (Fig. S1b). The smaller temperature biases can be related to smaller biases generally found also for radiation and surface heat fluxes in the new configuration (Fig. S2, regions defined in Lindstedt et al. (2015)).

## 3 Model performance

The three configurations of HCLIM38 have been used over several different regions and climates. Since HCLIM38-ALADIN is a limited-area version of the global model ARPEGE the expectation is that it can be used in principle anywhere on Earth. Consequently, the model is not tuned for specific regions. HCLIM38-ALARO has mostly been used in high latitudes where it performs well. HCLIM38-AROME has been successfully used on domains ranging from the tropics and different mid-latitude regions to the Arctic, indicating that it can be used for various climates without additional modifications. Here we illustrate the
capability and performance of HCLIM38 with a number of selected examples for different domains and model configurations. More in-depth evaluation studies are left for separate papers for each of the domains analysed below. The model experiments are summarized in Table S1.

### 3.1 Performance over European domains

#### 3.1.1 Pan-Europe

Here, HCLIM38 is compared to other state-of-the-art RCMs over Europe (Kotlarski et al., 2014). HCLIM38-ALADIN has been run with a couple of domain configurations similar to the EURO-CORDEX domain with the horizontal grid spacing of





12 km (EURO-CORDEX uses 0.11° resolution; Kotlarski et al. (2014)). The boundary conditions were taken from the ERA-Interim reanalysis (Dee et al., 2011). Differences in daily mean near-surface air temperature (T2m) and precipitation between nine RCMs (including HCLIM38-ALADIN) and EOBS version 17 gridded observations (Haylock et al., 2008) were calculated for the PRUDENCE regions in Europe (Christensen and Christensen, 2007) based on a common 10-year period (1999–2008).

RCMs were interpolated to the EOBS 0.25° regular grid prior to comparison. Figure 2 shows the results for winter (DJF) and summer (JJA) seasons for the Scandinavia and Mediterranean regions. The results for these two regions, which are representative for most PRUDENCE regions, show that HCLIM38-ALADIN is generally colder and wetter than EOBS. However, it can be seen that most of the other RCMs are wetter than EOBS too and that HCLIM38-ALADIN is generally in close agreement with EOBS in winter (Fig. S1). Larger differences are mostly seen in connection with mountainous regions. This is also the

case for temperature in winter. For example, the negative temperature bias in winter over Scandinavia is mostly associated with 2–4 °C lower values over the Scandinavian mountains. At lower altitudes the temperature is much better represented by the model (Fig. S1). Still, the cold bias in HCLIM38-ALADIN is present throughout most seasons and is spatially widespread over Europe. The reason for this systematic bias is not yet known and needs to be further investigated. Finally, we note that for the two versions of ALADIN, HCLIM38-ALADIN consistently shows lower temperatures than CNRM-ALADIN53 for

large parts of Europe, apart from western and southern Europe. The precipitation differences compared to EOBS, however, are generally smaller in HCLIM38-ALADIN. We conclude that the performance of HCLIM38-ALADIN over Europe, in terms of multi-year means of T2m and precipitation, is generally within the range of the performance of other RCMs used within EURO-CORDEX.

### 3.1.2   Norway

Due to its complex topography and the exposure of the western coast to large amounts of moisture transported over the North Atlantic, Norway is a challenging area to provide accurate climate model simulations and constitutes an ideal testing environment. To evaluate the performance of the HCLIM38-AROME model in this region at convection permitting scales, it was run at 2.5 km for an area covering Norway, parts of Finland, Sweden and Russia for the time period 2004–2016 (Crespi et al., 2019), downscaling the ERA-Interim reanalysis. An intermediate HCLIM38-ALARO simulation at 24 km resolution

was used to avoid a too large jump across the boundaries (results not analysed here).

For the evaluation over Norway, we are using the gridded observation dataset seNorge v2.1 for precipitation (Lussana et al., 2018a) and temperature (Lussana et al., 2018b). seNorge provides daily precipitation and temperature back to 1957 at a grid resolution of 1 km. The dataset has a special focus on providing meteorological input for snow and hydrological simulations at the regional or national level. Thus it not only covers the Norwegian mainland but also neighbouring areas in Finland, Sweden

and Russia to include regions impacting the Norwegian water balance. The station data are retrieved from MET Norway's climate database and the European Climate Assessment and Dataset (ECA&D, Klein Tank et al. (2002)). More details on the spatial interpolation schemes and the dataset in general are given in Lussana et al. (2018a, b).

Note that gridded observation data sets in remote regions of Norway should be considered with caution. For certain regions of Norway it may be difficult to judge the quality of the seNorge data due to the inhomogeneous station distribution. While





terrain height can reach 2000 m and more, most stations are located below 1000 m. There is also a difference in the station density between the southern and the northern parts with a higher density in the south of Norway (Lussana et al., 2019). For precipitation, Lussana et al. (2018a) have shown that seNorge2 gives too low values in very data-sparse areas. For temperature, an evaluation in Lussana et al. (2018b) has shown that the interpolation actually yields unbiased estimates, with the exception

of a warm bias for very low temperatures, and that the grid point estimates show a precision of 0.8 to 2.4 °C.

Generally, HCLIM38-AROME shows an underestimation of precipitation at the southwestern coast and over the Lofoten islands, while precipitation in the mountains is overestimated (Fig. 3). The largest differences appear in autumn and winter (not shown). Too high values for precipitation can also be seen for spring and summer over central-southern Norway. Similar differences have been shown by Müller et al. (2017) for the Nordic operational NWP setup of HARMONIE-AROME and by

Crespi et al. (2019) comparing the HCLIM38-AROME precipitation data to observed precipitation.

However, despite the biases found in simulated precipitation, the HCLIM38-AROME data has been successfully used by Crespi et al. (2019) together with in situ observations to obtain improved monthly precipitation climatologies over Norway. The high-resolution model data was used to overcome the uneven station network over Norway and it was shown that the simulated precipitation could be used to improve the climatologies significantly, especially over the most remote regions. For

instance, for the wettest area in Norway around Ålfotbreen, the combined data results in a mean annual precipitation of over 5700 mm. Measurements of snow accumulation during the winter half-year together with estimates of drainage from river basins (performed by the Norwegian Water Resources and Energy Directorate) indicate mean annual precipitation amounts over 5500 mm in the area (Teigen, 2005). The purely observation based estimates are almost 2000 mm lower.

The combined dataset shows systematically higher precipitation values than seNorge for the observation-sparse mountainous

regions. As the seNorge dataset is likely to underestimate precipitation in these areas (Lussana et al., 2018a) we expect this to be an improvement. However, whether this is true still needs to be verified, e.g. by the use of hydrological simulations (Lussana et al., 2019).

Annual mean temperature in HCLIM38-AROME is generally lower than in seNorge but the differences are small (Fig. 4). Averaged over the whole area, HCLIM38-AROME is about 1 °C colder than seNorge. The differences are ranging from about

-7 °C to 4 °C but for most of the area, the temperature differences are below ±2 °C and larger biases are restricted to the mountains. All seasons show similar patterns to the annual biases with the largest differences in the mountains, but the general increase of the bias with altitude is most pronounced in winter (not shown).

### 3.1.3 Summer precipitation over the Netherlands and Germany

Better resolving summertime precipitation is arguably the most important reason for running CPRCMs, and a primary variable

for which we anticipate added value compared to coarser resolution RCM models. In this section we discuss statistics of summer precipitation in two 10-year ERA-Interim forced climate simulations for the period 2000–2009, carried out with HCLIM38-AROME at convection permitting scales. Focus will be on the summer half year (April–September). The first CPRCM experiment (E1), conducted by KNMI, receives lateral boundary conditions from the RCM RACMO2 (van Meijgaard et al., 2012). In the second experiment (E2), conducted by SMHI, the host model is HCLIM38-ALADIN. Both host models





are forced by ERA-Interim. For both E1 and E2 the domain covers the pan-Alpine region as defined in the CORDEX FPS on convection (Coppola et al., 2018), but the E1-domain extends substantially further to the north-west covering a large part of the North Sea. Output is compared to two types of observations: radar data (for the Netherlands) and rain gauges (for a part of Germany).

For the evaluation of the CPRCMs to rain radar over the Netherlands, we can only use output from E1 since the E2 domain does not cover the Netherlands. The radar data is an hourly gridded product (2.4 km horizontal resolution) that has been calibrated against automatic and manual rain gauges (Overeem et al., 2009). Data of both the CPRCM (HCLIM38-AROME, 2.5 km) and the RCM (RACMO2, 0.11°) are put on the radar grid using nearest-neighbor interpolation. Two statistics are studied. First we look at hourly spatial precipitation maxima found within the Netherlands. We call this the hourly FLDMAX

statistics. The analysis focuses on model performance at the grid scale, and here we expect to find added value. Note that in this formulation the FLDMAX statistics does not account for the spatial extent of the precipitation. Neither does it account for the possible existence of several convective clusters at the same hour. Secondly, we study the hourly FLDMEAN statistics (i.e., the hourly area-average precipitation). If a CPRCM outperforms its host model for the latter statistics, this is an example of up-scale added value: the higher horizontal resolution pays off also at larger spatial scale. This is not guaranteed, especially

not in winter, when precipitation is often caused by large-scale weather systems. Confidence intervals are estimated using bootstrapping. This bootstrapping is implemented as sampling with replacement from the hourly data. The resulting estimate is overconfident as it ignores time correlations.

We expect the largest benefits of using a high-resolution model at the finest scales, which cannot be reached by the non-CPRCM. This is confirmed for the Netherlands in Fig. 5a, which shows for each hour an estimate of the average and the 90th

and 99th percentile of the FLDMAX precipitation. Three-hourly sliding windows are used to improve statistical robustness and confidence bands were determined using bootstrap resampling on hourly basis. HCLIM38-AROME simulates the diurnal cycle very well. Although the diurnal cycle of HCLIM38-AROME is about an hour delayed with respect to the radar observations, it is much improved compared to RACMO2 which hardly shows signs of a diurnal cycle. Results for the winter half year are qualitatively similar but the amplitude of the daily cycle is much less pronounced (not shown). For the FLDMEAN statistic,

the amplitude in diurnal cycle is dampened and RACMO2 and HCLIM38-AROME are generally more similar (not shown). Figure 5b and c shows the precipitation distributions as exceedance plots. Not only at the grid-scale (FLDMAX), but also at the largest spatial scale available (FLDMEAN, the Netherlands), HCLIM38-AROME generally outperforms RACMO2, especially for the larger precipitation amounts.

For the evaluation of the CPRCMs over Germany, we compare E1 and E2 to hourly rain gauge data. As a proxy for the

rain gauge location we use the nearest model grid point. In addition, we consider only stations that lie below 500 m altitude (see Fig. S3 for the rain gauges involved). Similarly to evaluation over the Netherlands we use the FLDMAX and FLDMEAN approach in which the appropriate statistical operator is applied over the hourly data.

The differences between CPRCM and RCMs for FLDMAX for Germany are similar to the results for the Netherlands presented above, with the CPRCM agreeing better with observations (Fig. 6). Note how similar the HCLIM38-AROME sim-

ulations are, despite the rather different characteristics of their host model. Although the diurnal cycle is better represented by





the CPRCMs, the modelled late afternoon peaks are too high. It appears that the overestimation of the peak is related to the elevation: If we constrain to the subset spanned by gauges at lower elevation, the CPRCMs agree better with the observations (not shown). Similar to the results over the Netherlands, the differences between CPRCM and RCM are smaller for the FLD-MEAN statistic (not shown). In addition, if we pool the data, rather than aggregate it, the results are qualitatively similar to the
FLDMAX statistic (not shown).

### 3.1.4   Summer precipitation over the Iberian Peninsula

The Mediterranean region is characterized by a complex morphology, due to the presence of many sharp orographic features: high mountain ridges surround the Mediterranean Sea on almost every side, presence of distinct basins and gulfs, islands and peninsulas of various sizes. These characteristics have important consequences on both sea and atmospheric circulations,
because they introduce large spatial variability and the presence of many sub-regional and mesoscale features (e.g., Lionello et al., 2006). Moreover, the Mediterranean region is considered to be a hotspot for climate change (e.g., Giorgi, 2006). It is frequently exposed to recurring droughts and torrential rainfall events, both of which are projected to become more frequent and/or intense as a result of future climate change. Therefore it is an area where climate models have to be tested. In this study we focus on the eastern Spain in summer (JJA) when precipitation events are scarce and mainly convective (e.g., Alhammoud
et al., 2014). Our aim is to study the performance of HCLIM38 in representing convective precipitation in dry zones at high temporal and spatial resolution.

We compare results from two HCLIM38 simulations, one at high resolution with AROME physics and another at lower resolution with ALADIN physics, against hourly precipitation records. These are the experiments performed:

   – AROME: 10 years (1990–1999), 2.5 km resolution, Iberian Peninsula

– ALADIN: 10 years (2005–2014), 12 km resolution, Europe

Both simulations were forced directly with ERA-Interim data at the boundaries. The simulations are compared with a dense set of around 500 automatic and manual rain gauges recording hourly precipitation that are distributed over Eastern Spain (Fig. S4). These observations are extracted from the National Weather Data Bank of AEMET (Santos-Burguete, 2018, Chapter 9) and have been used in other studies of intense precipitation events (e.g., Khodayar et al., 2016; Riesco-Martín et al., 2014).
The selected period for observations is 2008–2018 because hourly data from automatic stations started to be available in 2008. This period has seven years of overlap with the HCLIM38-ALADIN simulation but it does not overlap with the HCLIM38-AROME simulation. We assume that the precipitation characteristics have not substantially changed in the last 30 years and that in two near 10 year periods the mean statistics calculated from hourly precipitation are comparable. The assumption that rainfall characteristics are not dependent on the period analysed, when periods are not far from each other, will
be verified below and has been shown in other studies (e.g., Brown et al., 2010).

Two aspects of summer (JJA) precipitation are studied: the timing of the maximum hourly precipitation within a day and the hourly intensity. At hourly scales we expect that the convection permitting non-hydrostatic physics (AROME) better reproduces the precipitation characteristics compared to the hydrostatic physics (ALADIN). For the timing and intensity we calculate for





each rain gauge the 10-year mean of the hour of maximum precipitation in a day and the corresponding hourly intensity, respectively. The same is calculated for the two models after interpolating the modelled precipitation to the nearest neighbour observation. The 10-year observations mean is subtracted from the 10-year model means to get the difference in every point. These differences for every location are pooled together and probability distribution functions (PDF) of the timing differences

and hourly intensity differences are obtained for AROME and ALADIN. Gaussian distributions are fitted to the PDFs with the same mean and variance for a clearer comparison.

Several hourly thresholds have been tested (0, 5, 10, 15 and 20 mm h$^{-1}$), filtering out hourly precipitation below each threshold. Finally, we use the threshold of 5 mm h$^{-1}$ to remove periods with weak precipitation, while keeping sufficiently many events for robust statistics. The number of points fulfilling this condition is near 500 for AROME and around 400 for

ALADIN.

The histograms of the differences of the hour of maximum precipitation and intensity between models and observations are shown in Fig. 7. The distributions of differences for both timing and intensity are centered at zero for HCLIM38-AROME, while they are shifted to negative values for HCLIM38-ALADIN. This indicates that the ALADIN physics underestimates both the timing and intensity of maximum hourly precipitation, while the convection permitting physics AROME is able to

reproduce both realistically. In the case of timing the PDF of HCLIM38-AROME is narrower while HCLIM38-ALADIN shows a wider spread in the hour of maximum precipitation. For the intensity both models have similar spread.

Even though HCLIM38-ALADIN has seven years of overlap with observations and HCLIM38-AROME has none, HCLIM38-AROME is closer to the observations. This indicates that observations and simulations of different (but close) periods can be compared in this case.

In conclusion, the convection permitting model HCLIM38-AROME shows a clear improvement in the representation of key characteristics of precipitation as exemplified by timing and hourly intensity compared to the hydrostatic model HCLIM38-ALADIN. While HCLIM38-ALADIN underestimates the mean values of both variables, HCLIM38-AROME shows very small biases for the used threshold (> 5 mm h$^{-1}$). In accordance with previous CPRCM studies (e.g., Ban et al., 2014; Berthou et al., 2018), the main reason of this improvement is that AROME explicitly resolves deep convection (Seity et al., 2011; Bengtsson

et al., 2017) while HCLIM38-ALADIN parameterizes it.

### 3.2 First simulations over the Arctic region

The Arctic region is an excellent testbed for climate models. Interactions between the atmosphere, ocean and cryosphere are central in shaping the regional climate, and biases in the simulated climate will be greatly affected by the model's ability to capture the correct build-up and melt of snow and ice. Previous studies have indicated the challenges in simulating the

Arctic climate, and highlighted that more detailed surface schemes and very high resolution may be needed to improve model performance in the Arctic and on Greenland (e.g., Ettema et al., 2010; Lucas-Picher et al., 2012; Rae et al., 2012; Noël et al., 2018).

HCLIM38 can run in a polar stereographic projection which is ideal for the Arctic region. Here, we present the results from three HCLIM38 simulations; all forced by ERA-Interim during the summer of 2014 over a domain covering the Arctic





region reaching 60°N across all longitudes. The simulations all use the default HCLIM38 setup, and do not employ any type of nudging despite the large domain. The three experiments are:

- ALARO24: HCLIM38-ALARO, 24 km resolution

- ALADIN24: HCLIM38-ALADIN, 24 km resolution

- ALADIN12: HCLIM38-ALADIN, 12 km resolution

This ensemble allows for assessment of (1) performance of HCLIM38 compared to observed conditions, (2) the differences between identical simulations with ALARO and ALADIN physics and (3) the impact of increased resolution (in ALADIN only).

Given that this large domain is only forced by data on the lateral boundaries and at the sea surface, we do not expect matching
daily and sub-daily variability. Hence, the following assessment is based on integrated monthly and seasonal values rather than day-to-day comparison between model and observations.

The model performance is assessed over Greenland using in situ observations at 16 locations from the Programme for Monitoring of the Greenland Ice Sheet (PROMICE; van As et al. (2011); Fausto and van As (2019)); a network of automatic weather stations placed on the Greenland ice sheet. We assess the monthly and seasonal mean temperature in the model grid
cells closest to the stations, which are mainly located in the ablation zone; i.e. the lower elevation part of the ice sheet, which experiences melt during summer. To ensure a consistent bias assessment, the simulated temperatures have been corrected for the elevation difference between the station location and the model grid cell. Figure 8a shows seasonal (JJA) mean temperature in the model and the bias at the individual PROMICE stations, where the simulated temperature is adjusted using a lapse rate of 6.0 °C km$^{-1}$ (based on observed summer conditions; Erokhina et al. (2017)). Further, we have calculated the number of
days in which these locations experience melt, here defined as near-surface air temperature above 0 °C. This is evaluated using the Tmax, i.e. the daily maximum temperature experienced in a given grid cell on time step scale (time steps are 10 min for ALARO24 and ALADIN24, and 5 min for ALADIN12). Here the simulated temperature is not lapse rate corrected; melt is chosen as an additional metric in order to assess the model representation of the cryospheric feedback processes that are central to the Arctic climate.

Compared to the driving ERA-Interim reanalysis data, all the HCLIM38 runs provide more detailed spatial patterns over complex terrain; in mountainous regions and on the slopes of the ice sheet. The comparison to the PROMICE observations, which are all located on the slopes, reveals that the model generally performs better on the more elevated sites compared to the lower sites at the same locations. The southernmost stations (QAS_U and QAS_L) both have larger biases in HCLIM38 compared to the coarser ERA-Interim (approximately 80 km resolution). Notably, the upper station (QAS_U) has a warm bias
while the lower (QAS_L) has a cold bias. As previously shown (e.g., Lucas-Picher et al., 2012), this southern region has a very complex terrain with high peaks and deep fjords, which causes large variability between neighboring grid cells making the evaluation of model runs on scales like these difficult. As evident from Fig. 8b, it is worth noting that despite the biases, all three model versions achieve the correct number of days with melt at both the upper and lower stations. Another consistent





feature across the models is a cold bias at the SCO stations (SCO_U and SCO_L) on the central east coast. The difference between the altitude in the model and the two stations is very large; indicating that the model (even at 12 km resolution) does not describe the complex terrain, and thus the local circulation, adequately. Hence, a notable bias remains even after correcting for the elevation difference at the station location.

The melting season in all three models exhibits a general pattern of positive biases with too many days of melting in June and negative biases with too few days in August; i.e. a shift towards the earlier part of the summer. The northernmost sites (KPC and THU), especially, appear to have a too early start of the melting season in the models.

The two 24 km experiments have a very similar spatial pattern of biases across the stations; the main difference being that ALARO24 is overall warmer than ALADIN24, resulting in increased warm biases and decreased cold biases. The warmer

conditions in ALARO24 are also evident in north and northeastern Greenland on the central, more elevated parts of the ice sheet. Remarkably, the mean temperature in July even exceeds the melting point in a widespread area (not shown; but the pattern is reflected in the JJA mean temperature in Fig. 8a). This appears excessive compared to observations, where widespread melt on the interior ice sheet has only occurred twice over the past several centuries (e.g., Nghiem et al., 2012). The most recent occurrence in July 2012 was related to exceptional conditions in the atmospheric circulation (Neff et al., 2014; Fausto et al.,

2016). While 2014 did have one of the highest melt extents observed, the central, most elevated parts of the ice sheet did not experience melt (Tedesco et al., 2015). The spatial pattern of melt in ALADIN24 and ALADIN12 thus appear more consistent with the observed conditions in 2014. Considering the amount of melt days at the PROMICE stations (Fig. 8b), the warmer conditions in ALARO24 result in slightly better agreement with observations on average (by increasing melt at the stations with a cold bias).

The increased resolution in ALADIN12 results in overall colder conditions on the slopes of the ice sheet. The doubled resolution is able to resolve the steep slopes better, and consequently the along-slope temperature gradient on the ice sheet is increased. In terms of melt days, the average total number of melt days over the summer is very close to observations in ALADIN12.

Previous work by Mottram et al. (2017) reveals that HARMONIE-AROME (i.e. the NWP version of the convection permit-

ting model used in HCLIM) suffers from potential errors related to loss of the snow cover over glacier surfaces. When the snow cover is lost in their model setup, the exposed surface is not classified as ice and can, unphysically, heat up above the freezing point, distorting temperature and atmospheric circulation patterns compared to observations. None of the three experiments examined here show signs of complete loss of snow cover or the related issues reported earlier. Assessing whether this is a result of different model physics, coarser resolution, or the updated surface scheme employed in our HCLIM38 simulations

requires further investigation.

Together, these results indicate reasonable performance of HCLIM38 in the Arctic melt season. While all three model setups tend to have the peak melting season occurring too early, the overall length of the melt season corresponds to the observed. The results do, however, highlight the importance of very high resolutions for capturing the conditions on the coastal slopes of Greenland. The model-data comparison for the summer 2014 suggests a general cold bias over Greenland in HCLIM38, with

ALARO physics resulting in slightly warmer conditions compared to ALADIN. While the warmer conditions in ALARO24


improves the average agreement with PROMICE observations in the ablation zone of the ice sheet, the additional warming on the north-northeastern part of the central ice sheet appears excessive compared to observations.

### 3.3 Simulations over Africa

In the Future Resilience for African CiTies And Lands project - FRACTAL (http://www.fractal.org.za) HCLIM38 has been
used for a study on added value of dynamical downscaling over Africa. HCLIM38-ALADIN has been used for downscaling the ERA-Interim reanalysis over the CORDEX-Africa domain with four grid spacings: 25, 50, 100 and 200 km. Additionally, within ELVIC CORDEX FPS, HCLIM38-AROME has been used to generate 2.5 km simulations over the Lake Victoria Basin using the ERA-Interim forcing and HCLIM38-ALADIN as an intermediate model.

#### 3.3.1 HCLIM38-ALADIN performance over Africa

Since HCLIM38 has not previously been applied over the African continent, the ALADIN pan-African simulation at 50 km was compared to the CORDEX-Africa (http://www.csag.uct.ac.za/cordex-africa) RCM ensemble at about the same resolution (Nikulin et al., 2012, 2018). Figure 9 shows an example with the differences in mean July-September near-surface temperature and precipitation over the West Africa/Sahel region between the RCMs and the Climate Research Unit Time-Series (CRU, v. 4.01; Harris et al. (2014)). HCLIM38-ALADIN is somewhat dry and cold but the performance is well within the range of that
of the CORDEX-Africa ensemble. On annual timescale HCLIM38-ALADIN biases are mainly between -2 °C and 1.5 °C for temperature and -1.5 mm day$^{-1}$ and 1.5 mm day$^{-1}$ for precipitation (Fig. S5) that is also within the range of the CORDEX-Africa ensemble.

#### 3.3.2 Convection permitting simulations over Lake Victoria

In order to define an optimal configuration for convection permitting HCLIM38-AROME simulations in Africa, a number of
sensitivity experiments were performed for 2005–2006 over a wider Lake Victoria region within the ELVIC CORDEX FPS. The experiments include downscaling of ERA-Interim by HCLIM38-ALADIN at 25 km (pan-African and eastern Africa) and 12 km (eastern Africa) resolution. In a subsequent step HCLIM38-AROME downscaled all three HCLIM38-ALADIN simulations over the Lake Victoria Basin at 2.5 km. In the HCLIM38-ALADIN simulations precipitation over land is triggered too early during the diurnal cycle compared to the Tropical Rainfall Measuring Mission (TRMM-3B42, v. 7; Huffman et al.
(2007)) (Fig. 10). This too early peak is a common problem in the majority of the CORDEX-Africa RCMs (Nikulin et al., 2012). In contrast to HCLIM38-ALADIN, the time of maximum precipitation intensity during the diurnal cycle over land is very well simulated by HCLIM38-AROME at 2.5 km. This is in accordance with the HCLIM38-AROME results shown above for different European regions and with previous results using other CPRCMs (Finney et al., 2019), and strongly implies that the improvement is a result of the difference in model physics, i.e. the absence of convection parameterization, rather than only
increased resolution.





## 4   Conclusions and outlook

HCLIM is a regional climate modelling system developed from NWP operational models. It is based on the collaboration between several European national meteorological institutes, with the main focus on the development of a CPRCM that can be used in the emerging field of convection permitting regional climate simulations. It is a relatively new modelling system for

climate applications, with the currently recommended version HCLIM38 being used in an increasing number of national and international projects. The system is developed through a series of steps, starting from the ECMWF IFS, through the ALADIN-HIRLAM operational NWP model configurations, and finally to the specific climate modelling system. As such it contains a number of possible model configurations whose terminology needs proper explanation. Hence this paper is both timely and needed for clarity and proper understanding of the details of the system.

HCLIM38 consists of the three main atmospheric physics packages: ALADIN, ALARO and AROME, originally intended to be used at different resolutions even though there is some overlap in the scales they can be applied for. They are all coupled to the same surface model SURFEX and use the same options for the surface. The current standard choice in HCLIM38 is ALADIN for grid spacing close to or larger than 10 km, and AROME for grid spacing less than 4 km. HCLIM38 has so far been used in a wide range of different climates, from equatorial to polar regions, and the initial analysis presented here

indicates that it performs satisfactorily in those conditions. We also see that there are considerable benefits of using a CPRCM, particularly for sub-daily intense precipitation. HCLIM38-AROME is able to realistically simulate both the diurnal cycle and maximum intensity of sub-daily precipitation, which coarser RCMs or GCMs generally cannot accomplish. Our results contribute to this conceptual shift in climate simulations that allows for better understanding and quantifying extremes in a changing climate. Further studies, which are underway or planned, will analyse and evaluate the modelling system for specific

domains and applications.

One of the goals of HCLIM development is to interact closely with and benefit from NWP activities in model development, particularly related to HARMONIE-AROME. There are two main causes that currently hinder direct collaboration: HCLIM lags in cycles behind NWP, and HCLIM uses more sophisticated surface and soil model parameterizations. However, in the next operational cycle, cycle 43, both of these hurdles will be greatly diminished, because the NWP surface setup is going to

be the same as for climate applications, and both the NWP and climate researchers are developing the same model cycle for the next operational version. This benefits both groups in a number of ways. The usually smaller climate groups can focus on climate-specific questions while the general development is done in a wider context of NWP collaboration. At the same time, climate research provides insight into long-term model biases and feedbacks usually related to physical parameterizations, which can sometimes be masked in data-assimilation driven NWP short-range forecasts.

One common topic for both NWP and climate model developments is the aerosol treatment. In a recent work by (Rontu et al., 2019) first steps towards an updated aerosol climatology based on the Copernicus Atmosphere Monitoring Service (CAMS, Copernicus 2019, Flemming et al. (2017)) were taken. The authors showed that using the structure of the existing aerosol climatology (Tegen et al., 1997), they were able to introduce the CAMS based climatology to HARMONIE and run simulations with both configurations. The use of CAMS climatology led to slight improvements in the overall performance of

HCLIM38. However, this coarse approach was just an initial step and the future developments will bring the full potential of the CAMS approach including size and aerosol inherent optical properties (IOP) for all available species. This approach also allows for the full usage of all aerosol species available from CAMS, opens the possibility to combine the treatment of aerosols in radiation and cloud microphysics parameterizations, and enables the climate side to use significantly more temporally varying

aerosol climatology.

Other ongoing and planned HARMONIE and HCLIM developments include coupling to a wave model, an ocean model and a river routing model using the OASIS coupler available in SURFEX (Voldoire et al., 2017), a dynamic sea ice scheme, and a new groundwater scheme.

*Code availability.* The ALADIN and HIRLAM consortia cooperate on the development of a shared system of model codes. The HCLIM

model configuration forms part of this shared ALADIN-HIRLAM system. According to the ALADIN-HIRLAM collaboration agreement, all members of the ALADIN and HIRLAM consortia are allowed to license the shared ALADIN-HIRLAM codes within their home country for non-commercial research. Access to the HCLIM codes can be obtained by contacting one of the member institutes of the HIRLAM consortium (see links on http://www.hirlam.org/index.php/hirlam-programme-53). The access will be subject to signing a standardized ALADIN-HIRLAM license agreement (http://www.hirlam.org/index.php/hirlam-programme-53/access-to-the-models). Some parts of the ALADIN-

HIRLAM codes can be obtained by non-members through specific licences, such as in OpenIFS (https://confluence.ecmwf.int/display/OIFS) and Open-SURFEX (https://www.umr-cnrm.fr/surfex). The full code access is provided to the journal editor for peer review through a research license.

*Author contributions.* DB leads the HCLIM consortium, and has initiated and collated the paper. Authors HdV–FW contributed to developing or running and evaluating the model, and writing of the paper. Authors UA–MW contributed either to the model development or to model

evaluation and analysis of the results. All authors commented on the paper.

*Competing interests.* The authors declare that they have no conflict of interest.

*Acknowledgements.* We thank Jeanette Onvlee-Hooimeijer for helpful suggestions about the manuscript. The ALADIN-HIRLAM system is developed and maintained by a large number of contributors at different institutions. We are grateful to all of them, and it is only through their continuous effort that the HCLIM development could have started. Computational and storage resources for the simulations have been

provided by the European Centre for Medium-Range Weather Forecasts (ECMWF) and the computing resource Bi provided by the Swedish National Infrastructure for Computing (SNIC) at the Swedish National Supercomputing Centre (NSC) at Linköping University. Data from the Programme for Monitoring of the Greenland Ice Sheet (PROMICE) and the Greenland Analogue Project (GAP) were provided by the Geological Survey of Denmark and Greenland (GEUS) at http://www.promice.dk. The German station data were provided by the INTENSE



project from the European Research Council (grant ERC-2013-CoG-617329). The European Union's Horizon 2020 EUCP project (grant number 776613) funded parts of the research in this paper.





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



**Table 1.** List of main acronyms related to the HCLIM system

| Acronym | Full name | Notes |
|---|---|---|
| HIRLAM | High Resolution Limited Area Model | Collaboration between 10 national meteorological services (Bengtsson et al., 2017). Also, the name of the limited area model which is being phased out and has been replaced by HARMONIE-AROME |
| ALADIN | Aire Limitée Adaptation Dynamique Développement International | Collaboration between 16 national meteorological services. Also, the name of the limited area model using ARPEGE physics (Termonia et al., 2018) |
| HARMONIE | HIRLAM–ALADIN Research on Mesoscale Operational NWP in Euromed | The HARMONIE NWP system consists of the HIRLAM-specific AROME model configuration (Bengtsson et al., 2017), together with a scripting system and set of tools for building, running and validation/verification of the HARMONIE-AROME model |
| ARPEGE | Action de Recherche Petite Échelle Grande Échelle | Global model developed at Météo-France (Courtier et al., 1991) |
| AROME | Applications of Research to Operations at Mesoscale | Convection-permitting model developed at Météo-France with HIRLAM contributions (Bengtsson et al., 2017; Termonia et al., 2018) |
| ALARO | ALadin-AROme | Limited-area model applicable also in the convection "grey zone" (Termonia et al., 2018) |
| SURFEX | Surface Externalisée | Surface scheme shared by all HCLIM model configurations (Masson et al., 2013) |



**Table 2.** SURFEX parameterizations used in HCLIM38

| Model component | Parameterization | References |
|---|---|---|
| Sea and ocean | Exchange Coefficients from Unified Multi-campaigns Estimates (ECUME) | Belamari (2005); Belamari and Pirani (2007) |
| Sea ice | Simple ICE (SICE) | Batrak et al. (2018) |
| Soil | ISBA-DIF explicit multilayer scheme (14 layers to 12 m depth) | Boone (2000); Decharme et al. (2011) |
| Vegetation and carbon related processes | Jarvis-based stomatal resistance. Soil-organic carbon for soil thermal and hydrological properties | Noilhan and Planton (1989); Decharme et al. (2016) |
| Subgrid hydrology | Subgrid runoff. Horton runoff | Dümenil and Todini (1992); Decharme and Douville (2006) |
| Snow | ISBA-ES explicit snow scheme (12 layers) | Boone and Etchevers (2001); Decharme et al. (2016) |
| Town | Town Energy Balance (TEB) | Masson (2000) |
| Inland water | Freshwater lake model (FLake) | Mironov et al. (2010) |





**Table 3.** Parameterizations and dynamics of the three model configurations as used in HCLIM38

| Parameterization and dynamics | ALADIN | ALARO | AROME |
|---|---|---|---|
| Dynamics | Hydrostatic (Temperton et al., 2001) | Hydrostatic (Temperton et al., 2001) | Nonhydrostatic (Bénard et al., 2010) |
| Radiation | RRTM_LW, SW6 (Mlawer et al., 1997; Iacono et al., 2008; Fouquart and Bonnel, 1980) | ACRANEB2 (Mašek et al., 2016; Geleyn et al., 2017) | RRTM_LW, SW6 (Mlawer et al., 1997; Iacono et al., 2008; Fouquart and Bonnel, 1980) |
| Turbulence | CBR (Cuxart et al., 2000); mixing length from Bougeault and Lacarrere (1989) | pTKE (Geleyn et al., 2006) | HARATU (Lenderink and Holtslag, 2004; Bengtsson et al., 2017) |
| Microphysics | Lopez (2002); Bouteloup et al. (2005) | Lopez (2002) | ICE3-OCND2 (Pinty and Jabouille, 1998; Müller et al., 2017) |
| Shallow convection | KFB (Bechtold et al., 2001; Bazile et al., 2012) | Pseudo shallow convection parameterization (Geleyn, 1987) | EDMFm (de Rooy and Siebesma, 2008; Bengtsson et al., 2017) |
| Deep convection | Bougeault (1985) | 3MT (Gerard et al., 2009) | - |
| Clouds | Smith (1990) | Xu and Randall (1996) | Bechtold et al. (1995) |
| Orographic wave drag | Catry et al. (2008) | Catry et al. (2008) | - |




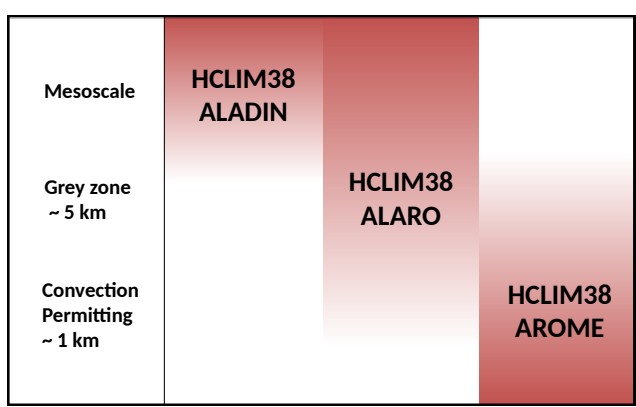

**Figure 1.** Intended horizontal grid resolutions for the three model configurations available in HCLIM38 (based on Termonia et al. (2018)).

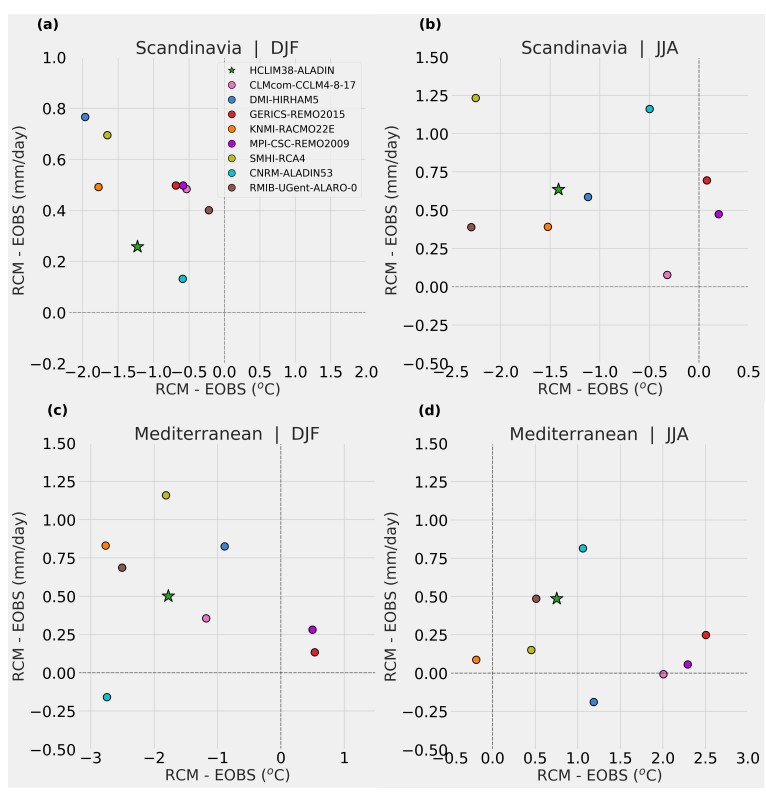

**Figure 2.** Difference between RCMs and EOBS of near-surface temperature (x axis) and daily precipitation (y axis) for (a), (c) DJF and (b), (d) JJA averaged over 1999–2008 for two PRUDENCE European sub-regions: (a), (b) Scandinavia and (c), (d) the Mediterranean.

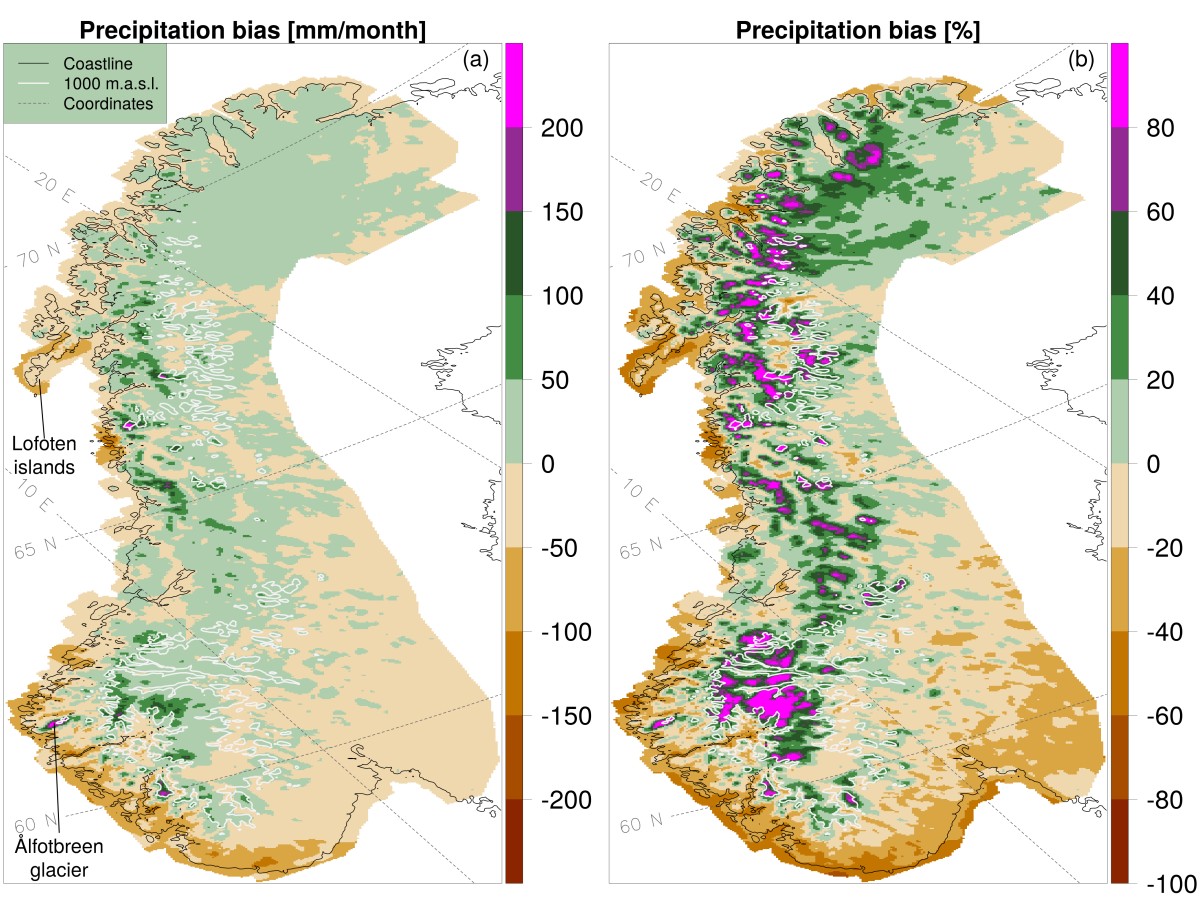

**Figure 3.** (a) Absolute and (b) relative precipitation differences between HCLIM38-AROME and seNorge for the time period 2004–2015.

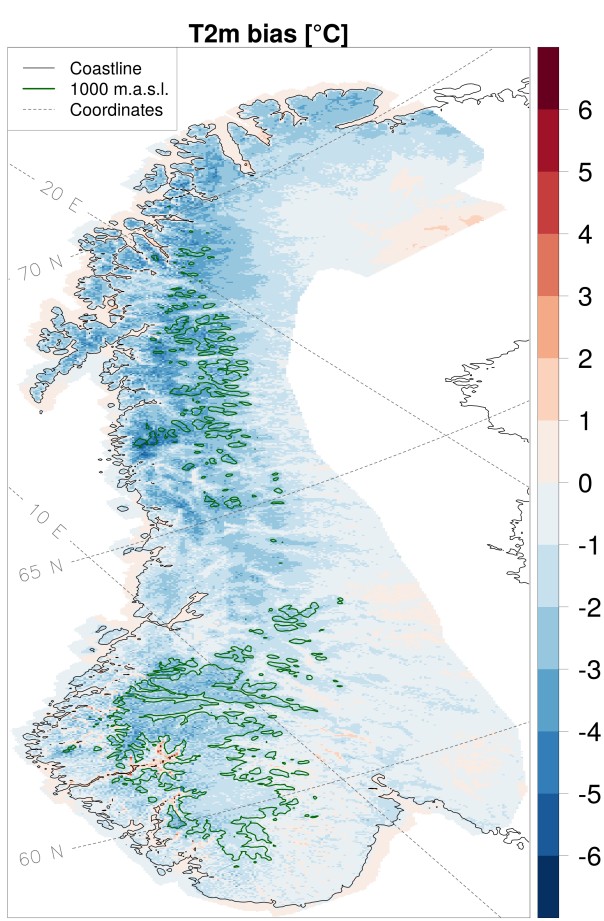

**Figure 4.** Annual temperature differences between HCLIM38-AROME and seNorge for the time period 2004–2015.





**(a)** Apr–Sep diurnal cycle for fldmax(precip)

**(b)** Apr–Sep exceedance statistics for fldmax(precip) **(c)** Apr–Sep exceedance statistics for fldmean(precip)

**Figure 5.** (a) Diurnal cycle of two high percentiles and the average of the FLDMAX hourly precipitation distribution (Apr–Sep), for radar, CPRCM and driving RCM. Probability of exceedance (Apr–Sep) for (b) the FLDMAX precipitation and (c) the FLDMEAN precipitation (note the difference in the vertical scale).



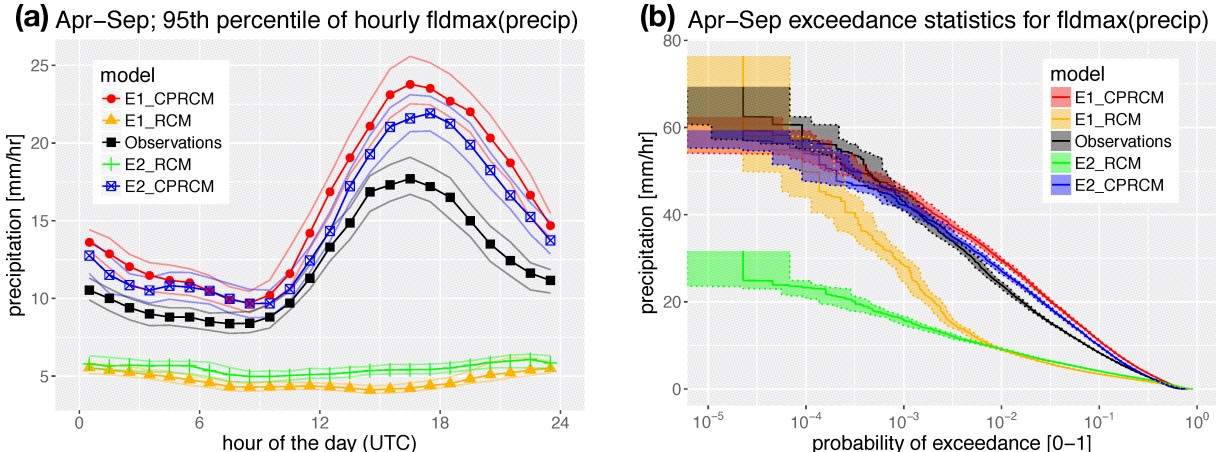

**Figure 6.** As in Fig. 5, but for the evaluation over low-lying (< 500 m) German rain gauges. (a) Diurnal cycle of the 95th percentile of the FLDMAX precipitation. (b) Probability of exceedance (Apr–Sep) for the FLDMAX precipitation.





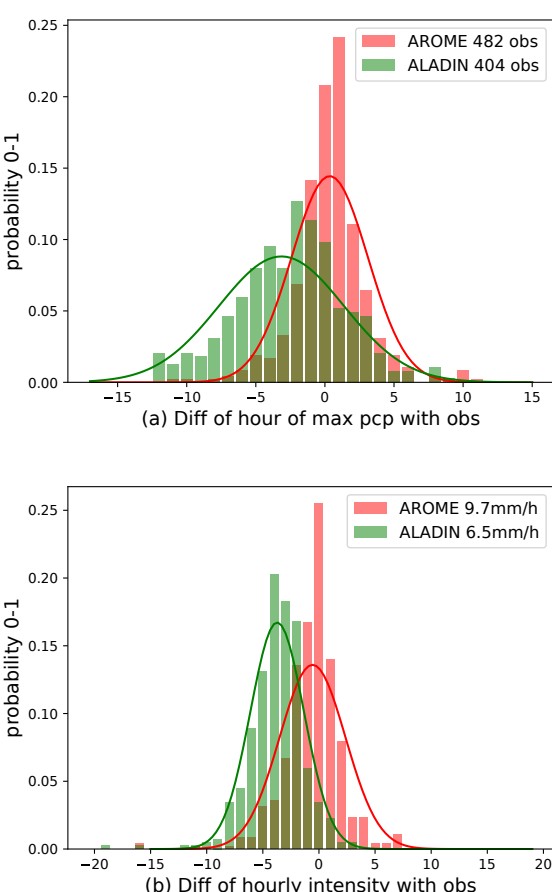

**Figure 7.** Histograms of differences between models (HCLIM-AROME 2.5 km and HCLIM-ALADIN 12 km) and observations of the 10-year mean of (a) hour of maximum precipitation and (b) the corresponding maximum hourly intensity for the hourly intensity threshold of 5 mm h$^{-1}$. The legend shows (a) the number of stations used in the comparison for each of the models and (b) the mean intensity for each model. The mean intensity in observations is 10.2 mm h$^{-1}$.



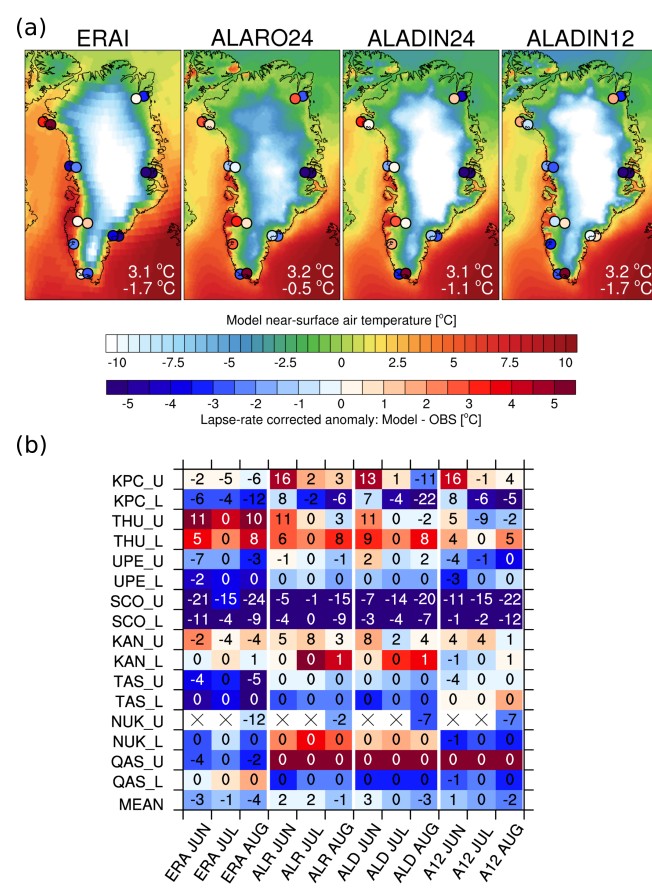

**Figure 8.** (a) The summer (JJA) mean temperature from ERA-Interim compared to the three HCLIM38 experiments. Colored dots indicate the anomalies between the lapse rate corrected model temperature and the observation at each PROMICE station. Numbers in the lower right corner indicate the root mean square error (top) and the average anomaly across the stations (bottom). (b) The monthly mean difference between the simulated lapse rate corrected temperature and the observations at each PROMICE station. The shading corresponds to the anomaly shading in (a), while the number indicates the anomaly in the number of melt days between model and observations (ordered by latitude, north to south). Negative (positive) numbers indicate that the model has too few (too many) melt days compared to observations. Crosses on white background indicate missing observational data. Model names have been shortened: ERA is ERA-Interim, ALR is ALARO24, ALD is ALADIN24, and A12 is ALADIN12.





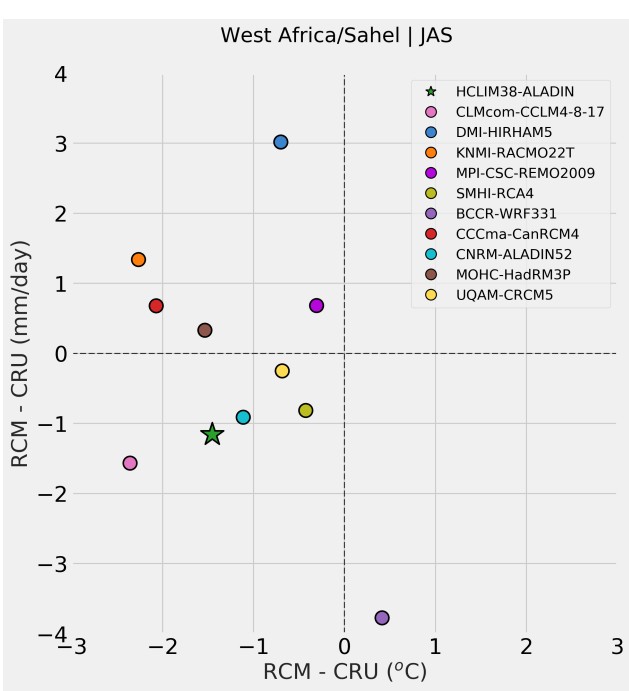

**Figure 9.** RCM biases (with respect to CRU) in mean July–September (2000–2009) temperature and precipitation over West Africa (7.5–15N, 10W–10E). Circles represent individual RCMs from the CORDEX-Africa ensemble and the star is HCLIM38-ALADIN. The horizontal resolution is 0.44° or 50 km.



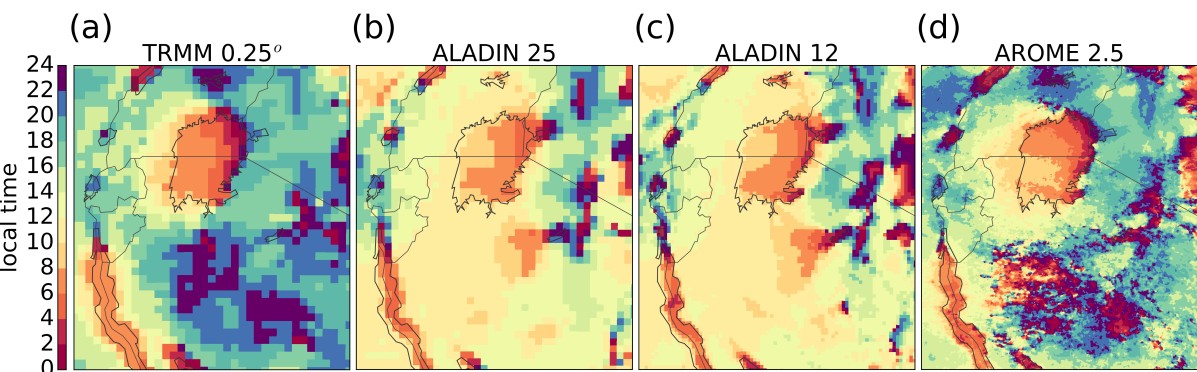

**Figure 10.** Time (local mean time) of maximum precipitation intensity during the diurnal cycle around Lake Victoria (2006) for (a) TRMM with 0.11° resolution, (b) ALADIN with 25 km resolution, (c) ALADIN with 12 km resolution and (d) AROME with 2.5 km resolution. Time of maximum precipitation is based on a cubic spline fitted to the 3 h precipitation and only days with more than 1 mm day$^{-1}$ are included (see Nikulin et al. (2012) for details). The AROME simulation was forced by the ALADIN 12 km simulation shown in (c).