# Peer review of "HCLIM38: A flexible regional climate model applicable for different climate zones from coarse to convection permitting scales"

_Geoscientific Model Development, 2019_

## Short Comment (SC1) · 16 Sep 2019

Dear authors, dear editor,

A short comment to update you with the latest references concerning the use of AL-ADIN and AROME in climate mode at CNRM. So probably some of those references can fit in your section 2.3. I let you decide what is relevant.

- I think that it is worse mentioning that ALADIN has been used in climate mode for more than 10 years (see for example, Radu et al. 2008, Déqué and Somot 2008, Colin et al. 2010) in particular in projects such as ENSEMBLES or CORDEX.

[Figure]

- Before being used in cycle 41t1 (Coppola et al. 2019), AROME has been used in climate mode at CNRM with the cycle 38 (same cycle you are describing in this GMDD) to explore the added-value of such CPRCM with respect to a twin 12km ALADIN simulation. References are: Déqué et al. 2016, Fumière et al. 2019

- There are references for the use of ALADIN in a fully coupled framework, the so-called RCSM configuration including the sea and river representation. For example, Sevault et al. 2014 (CNRM-RCSM4) and Darmaraki et al. 2019 (CNRM-RCSM6 the latest version based on ALADIN v6.3).

thank you in advance to take into account this comment

Samuel Somot

references:

Darmaraki S., Somot S., Sevault F., Nabat P., (2019) Past Variability of Mediterranean Sea Marine Heatwaves. GRL, doi:10.1029/2019GL082933

Fumière Q., Déqué M., Nuissier O., Somot S., Alias A., Caillaud C., Laurantin O., Seity Y. (2019) Extreme rainfall in Mediterranean France during the fall: added-value of the CNRM-AROME Convection-Permitting Regional Climate Model. Clim. Dyn, 1-15, doi:10.1007/s00382-019-04898-8

Colin J., Déqué M., Radu R., Somot S. (2010) Sensitivity study of heavy precipitations in Limited Area Model climate simulation: influence of the size of the domain and the use of the spectral nudging technique. Tellus-A, 62(5), 591-604, doi: 10.1111/j.1600-0870.2010.00467.x

Déqué M. and Somot S. (2008) Extreme precipitation and high resolution with Aladin. Idöjaras Quaterly Journal of the Hungarian Meteorological Service, 112(3-4):179-190

Radu, R., Déqué, M., Somot, S. (2008) Spectral nudging in a spectral regional climate model. Tellus. 60A(5):885-897. doi: 10.1111/j.1600-0870.2008.00343.x

[Figure]

Coppola E., Sobolowski S., Pichelli E., Raffaele F., Ahrens B., Anders I., Ban N., Bastin S., Belda M., Belusic D., Caldas-Alvarez A., Margarida Cardoso R., Davolio S., Dobler A., Fernandez J., Fita Borrell L., Fumiere Q., Giorgi F., Goergen K., Guettler I., Halenka T., Heinzeller D., Hodnebrog Ø, Jacob D., Kartsios S., Katragkou E., Kendon E., Khodayar S., Kunstmann H., Knist S., Lavín A., Lind P., Lorenz T., Maraun D., Marelle L., van Meijgaard E., Milovac J., Myhre G., Panitz H.-J., Piazza M., Raffa M., Raub T., Rockel B., Schär C., Sieck K., Soares P.M.M., Somot S., Lidija Srnec L., Stocchi P., Tölle M., Truhetz H., Vautard R., de Vries H., Warrach-Sagi K. (2019) A first-of-its-kind multi-model convection permitting ensemble for investigating convective phenomena over Europe and the Mediterranean. Clim Dyn, 1-32, doi:10.1007/s00382-018-4521-8 (published on-line)

Sevault F., Somot S., Alias A., Dubois C., Lebeaupin-Brossier C., Nabat P., Adloff F., Déqué M. and Decharme B. (2014) A fully coupled Mediterranean regional climate system model: design and evaluation of the ocean component for the 1980-2012 period. Tellus A, 66, 23967, http://dx.doi.org/10.3402/tellusa.v66.23967

Déqué M., Alias A., Somot S., Nuissier O. (2016) Climate change and extreme precipitation: the response by a convection-resolving model. Research activities in atmospheric and oceanic modelling. CAS/JSC Working group on numerical experimentation. Report No.46 (available at http://www.wcrp-climate.org/WGNE/blue_book.html) https://www.wcrp-climate.org/WGNE/BlueBook/2016/individual-articles/07_D%C3%A9qu%C3%A9_Michel_ClimateConvectionResolving.pdf

---

## Referee Comment (RC1) · Anonymous Referee #1 · 17 Sep 2019

This paper aims at explaining the set-up of HCLIM, a regional climate modelling system developed by a consortium of national meteorological institutes and based on the HARMONIE-AROME numerical weather prediction system. It also provides examples of its use and added value in different regional set-ups. I find the manuscript well-written and clear in terms of explanation of the model set-up. It is an impressive effort to gather pieces of work by different groups and institutes over different regions. However, I find it a bit disappointing that the metrics used between regions are very often completely different. The only exception is Germany and the Netherlands, and even then the percentiles used are not the same. It makes the comparison of performance between regions impossible, which is a shame given the use of a similar model setup. If some of the metrics, such as FLDMAX could be produced over all the regions where AROME is used, it would be great. Also, it would be nice to show whether the CPRCMs have decent mean precipitation/temperature in summer, as it has been shown that other CPRCMs have dry biases in summer due to different interactions between stronger rainfall rates and the soil scheme (Liu et al. 2017, Berthou et al. 2018). This would be important, as the inclusion of SURFEX in the climate set-up is key, and as you mention the inclusion of groundwater in the future, which actually solved the WRF warm/dry bias in the US (work by M. Barlage). This could be done by including a similar plot as Fig. 2 but showing ALADIN and AROME for the subregions analysed here (Norway, the Netherland, Germany, Eastern Spain), for example in the same plot.

Major comments:

- Regarding the comparison between RCM and CPRCMs, I find it unfair to compare ALADIN at 12km with station data. As mentioned in the Netherland part of the article, it is alright to compare FLDMEAN of the radar and two models, but not FLDMAX: by design, ALADIN produces 12kmx12km mean precipitation, not point data, so I think it is wrong to include ALADIN in the FLDMAX plots. However, it is fantastic that AROME reproduces this metric so well. If you want to include ALADIN in plots, just use FLDMEAN or FLDMAX on similar areas as ALADIN by aggregating the radar and AROME at its resolution. This is also true for the metric used in Spain.

- Fig. 6a: could you use 90th/99th percentile of FLDMAX to be consistent with Fig. 5a (or the other way round?)

- Fig. 7: again for the intensity metric, I would not use ALADIN in these plots, as it is unfair (see first comment). It would be so much easier for the reader to have the same metrics for Spain as for Germany and the Netherlands (also it takes a bit of time to understand this new metric!). Since you have hourly gauges available

[Figure]

there, I guess it would be possible? Would you also be able to find hourly gauges in Norway to also do the same plot?

- For Spain, why are you not using SON, which is much more convective than JJA? (e.g. like Fumiere et al. (2019). A comparison with their results for southern France could be interesting: they found that AROME has a very good distribution but underestimates the strongest and rarest rain rates.)

- In general, how are the simulations performing for lower precipitation values? (e.g. Berthou et al. (2018) found an underestimation of low-value precipitation in UKMO and to a lesser extend ETH-COSMO. These values are important to get a good climatology and often underestimated by convection-permitting models, which can also lead to soil moisture depletion in summer. It seems from Fig. 5c that there is a dry bias in the model?

Minor comments:

- P6 – lines 1-2: "this helps in decreasing...": the atmospheric models at 12 and 2.5km have very different rain rates (as you show later), so I doubt that the soil spin-up inherited from the 12km model actually helps: would you be able to show that your soil moisture is not drifting in the 2.5km simulations? You don't actually mention how long a spin-up you use for the 2.5km models.

- It is worth mentioning that results from Fig. 6 (overestimation of intense precipitation (10-40mm/h) are consistent with the analysis of the UKMO/ETH-COSMO models in Berthou et al. (2018) for Germany.

- P10Lines15-17: which confidence interval are you using from the bootstrap resampling? Can you expand the method a bit?

- P12 Lines17-20: "This indicates that observations... -> This indicates that the ALADIN biases in the distribution are probably larger than inter-decadal variability in the distribution.

- P13, Lines 9-11: this is valid for all the simulations, move this to the start of section 3 (or remove it).

- P13, Lines 32-33: does this mean that the biases are in a temperature range where actually it does not impact melting, or that diurnal cycle changes balance mean monthly changes? Can you expand a bit on this?

Fumière, Q., Déqué, M., Nuissier, O., Somot, S., Alias, A., Caillaud, C., . . . Seity, Y. (2019). Extreme rainfall in Mediterranean France during the fall: added value of the CNRM-AROME Convection-Permitting Regional Climate Model. Climate Dynamics. https://doi.org/10.1007/s00382-019-04898-8

Liu, C., Ikeda, K., Rasmussen, R., Barlage, M., Newman, A. J., Prein, A. F., . . . Yates, D. (2016). Continental-scale convection-permitting modeling of the current and future climate of North America. Clim. Dyn., 1–25. https://doi.org/10.1007/s00382-016-3327-9

---

## Referee Comment (RC2) · Anonymous Referee #2 · 15 Oct 2019

General Comments:

The goal of this study is to introduce the HCLIM38 model, explain its origins, and describe the results from a number of simulations at different resolutions, using different physics packages. The authors provide ample evidence of performance over differing parts of the globe and over varying topography/climates. Overall, the simulation results show that HCLIM38 performs as well as or better (specifically at higher resolutions) than other RCM/CPRCMs.

While the scientific evidence is strong, work is needed to clarify the origins of HCLIM38 and its different configurations to reduce confusion. In addition, I would streamline

the discussion of small nuances of HCLIM38 as compared to many other related, but different NWP and GCM/RCM systems, since the latter aren't the focus of this paper. Focusing on the three configurations of HCLIM38 and the over-arching differences will improve clarity.

With sufficient improvement to the manuscript, publication in Geoscientific Model Development should be considered.

Major Comments:

1. Section 2.1 (and beyond) is very confusing. Even after reviewing Table 1, I'm unable to make the right connections for each system. If HARMONIE is not an actual model, is it just HIRLAM-ALADIN with a scripting system? It would be good to specify that HIRLAM-ALADIN is being phased out for AROME in the text (not in the table). Also, distinction between a model and its self-titled physics suite is necessary (AROME is a model, but has a physics suite with the same name). Line 3 states that HCLIM is based on ALADIN-HIRLAM, but line 8 says HARMONIE-AROME is the basis for HCLIM. A clear delineation between what is a true model, what is just a configuration of a model, what is a physics package, and how HCLIM38 fits into this picture is needed. I understand that a slew of changing acronyms is part of our work, but clarification of this section is necessary, and would go a long way toward reader comprehension.

2. It would be beneficial to have a table/plot showing how the three HCLIM38 configuration climate simulations were run. What did the domains look like (maybe provide a plot for each)? What were the time steps and required wall clock times to run the simulations? What were the LBCs used and from what models (some of this is already in the text)? How often were the LBCs updated?

Minor Comments:

1. Please expand upon/describe what "added value" is referring to in the first paragraph of the introduction. I assume this is referring to climate forecast accuracy, but it would

be good to have a better idea of what metrics the authors have in mind.

2. Please briefly summarize the over-arching/major changes/improvements compared to older versions of HCLIM in the introduction.

3. Is there a hydrostatic vs. non-hydrostatic namelist option in HCLIM that is invoked when using the different physics packages (AROME vs. ALADIN or ALARO)?

4. More information is necessary on what the term "cycle" represents. What systems correspond to the "NWP model configurations" that are referred to as cycles? Which MET services are running them in real-time?

5. The authors discuss different model resolution configurations in 2.2, but then switch to physics packages with the same names as the models halfway through the paragraph. The delineation between the model and associated physics should be clear. Is the "HARMONIE-AROME" model the "similar NWP system" referenced in line 26 on Page 3? If so, it should be clarified and tied into the previous sentence.

6. The authors state that there is no deep convection parameterization, so it can only be used at convection-permitting scales. Are there any scale-aware schemes that could handle unresolved convection below 4 km?

7. Please describe how Fig. 2 was created. What is EOBS? Are these mean temperature/precip differences averaged across the whole 10-year period?

8. It appears as though Crespi et al., 2019 used the in-situ observations to correct the bias of HCLIM38-AROME precipitation data over Norway in order to arrive at a more accurate precipitation climatology. It might be good to explain this detail.

9. There is no Fig S1 or Table S1, but they are referred to in the text.

10. Please briefly describe the exceedance plots and how you calculated them.

11. Any idea why HCLIM36-ALADIN would have early biases for both the time of maximum precipitation and maximum hourly intensity (Fig. 7)? Something in the convective

parameterization scheme?

12. How did the CAMS based aerosol climatology improve HCLIM38 if it wasn't coupled with radiation or microphysics parameterizations? If they weren't coupled, weren't they just passive tracers?

Introduction - Are there scale-aware physics parameterizations in GCMs, RCMs, and CPRCMS (as there are in NWP) that could handle increasing resolutions of climate modelling and could shut themselves off appropriately?

Introduction - Are the terms convection-"resolving" and convection-"permitting" interchangeable for CPRCMS?

Page 2, Line 18 – I might expand briefly upon why nesting from O(10km) to O(1km) is not possible, aside from the reference, since this is regularly done in NWP.

Page 3, Line 21 – Climate or NWP limited-area models? Page 3, Line 26-29 – ALADIN or ALARO physics, model, or both? This needs to be clear. Page 6, Line 2 – I assume it be used with GCMs and not just RCMs for LBCs? Page 8, Line 3-4 – EOBS and PRUDENCE are not defined. Page 6, Line 23 – I'm not sure what the "e.g." is doing here. Page 10, Line 31 – There is no Fig. S3 as referred to in the text. Page 13, Line 16 – There is no Fig. S5 as referred to in the text.

---

## Author Comment (AC1) · 3 Dec 2019

[Note: the responses are marked with R below each comment]

Response to RC1

This paper aims at explaining the set-up of HCLIM, a regional climate modelling system developed by a consortium of national meteorological institutes and based on the HARMONIE-AROME numerical weather prediction system. It also provides examples of its use and added value in different regional set-ups. I find the manuscript well-written and clear in terms of explanation of the model set-up. It is an impressive effort to gather pieces of work by different groups and institutes over different regions. However, I find it a bit disappointing that the metrics used between regions are very often completely different. The only exception is Germany and the Netherlands, and even then the percentiles used are not the same. It makes the comparison of performance between regions impossible, which is a shame given the use of a similar model setup. If some of the metrics, such as FLDMAX could be produced over all the regions where AROME is used, it would be great. Also, it would be nice to show whether the CPRCMs have decent mean precipitation/temperature in summer, as it has been shown that other CPRCMs have dry biases in summer due to different interactions between stronger rainfall rates and the soil scheme (Liu et al. 2017, Berthou et al. 2018). This would be important, as the inclusion of SURFEX in the climate set-up is key, and as you mention the inclusion of groundwater in the future, which actually solved the WRF warm/dry bias in the US (work by M. Barlage). This could be done by including a similar plot as Fig. 2 but showing ALADIN and AROME for the subregions analysed here (Norway, the Netherland, Germany, Eastern Spain), for example in the same plot.

R: We thank the Referee for thorough reading of the manuscript and constructive comments. Referee's concerns about using the different metrics are understandable. This predominantly results from the nature of the studies shown. The modelling setup was not aimed for intercomparison between the different domains. Despite the fact that all the groups use the same modelling system, the simulations for each domain were done independently by different groups and with quite different setups. We argue that it would be easier to compare different models over the same domain and with more similar modelling setup than it is to compare the same model with very different setups. For example, regarding the domains that the Referee mentions, in the Netherlands domain HCLIM38-AROME was forced with the RACMO23 model at the boundaries, for Norway it was forced with HCLIM38-ALARO, for Spain directly with ERA-Interim without an intermediate model and for Germany with ALADIN and RACMO23. Furthermore, the domain sizes and simulation periods for the different domains are not the same, while for the Spanish domain AROME and ALADIN are neither coupled nor

do they cover the same simulation period. This makes a comparison similar to Fig. 2 very difficult to perform and to interpret. Any discrepancies found in these kinds of plots would require additional analyses and explanation, which would need to be done for each domain (and, when applicable, for each intermediate model) separately. The latter is not feasible in a single paper. Our approach for this paper (with a note that this is a model description paper rather than a full evaluation paper) was to show snippets of the model performance over the different domains and climate zones. The different metrics were chosen to highlight aspects relevant for those regions, and are also driven by the availability of observations. We could have chosen to show a smaller number of domains with more detailed evaluation, but we felt that it was useful to indicate that the model can be generally applied in all regions. Furthermore, each of these and other domains will be thoroughly evaluated in separate papers, some of which are in preparation (e.g., Ban et al., 2019: evaluation and intercomparison of different CPRCMs, including HCLIM38, over several European domains - results from CORDEX FPS-Convection and H2020 EUCP projects; Lind et al., 2019: evaluation of HCLIM38 over the Nordic region), or have already been submitted (Wu et al., 2019) or published (Crespi et al., 2019; Toivonen et al., 2019). We now clearly indicate in several places in the text that we present only some aspects the model performance in this paper.

Major comments:

- Regarding the comparison between RCM and CPRCMs, I find it unfair to com- pare ALADIN at 12km with station data. As mentioned in the Netherland part of the article, it is alright to compare FLDMEAN of the radar and two models, but not FLDMAX: by design, ALADIN produces 12kmx12km mean precipitation, not point data, so I think it is wrong to include ALADIN in the FLDMAX plots. However, it is fantastic that AROME reproduces this metric so well. If you want to include ALADIN in plots, just use FLD-MEAN or FLDMAX on similar areas as ALADIN by aggregating the radar and AROME at its resolution. This is also true for the metric used in Spain.

R: We understand the Referee's argument. There are two ways of approaching this:

either to be fair to the coarse model and aggregate high resolution data to its resolution, or to show the best performance of each model, which for the higher resolution model includes the effects of both the physical processes and higher resolution. We think that for comparisons aiming at analysing the effects of different physical processes only, the models should be remapped to the same, coarsest grid. However, from a user perspective, the main interest is in the capability of the model to reproduce reality. For CPRCMs, showing that the reality is better reproduced is usually hindered by the lack of appropriate high-resolution observational data. Since we did not intend to compare physical processes in this paper, and at the same time we have the appropriate datasets at high resolution, we believe that it is justified to keep the current approach in the main body of the paper. However, we now include in the Supplementary material the upscaled version of Fig. 5 for the Netherlands (where both AROME and radar could be upscaled to the same ALADIN grid for consistency, unlike at other locations).

- Fig. 6a: could you use 90th/99th percentile of FLDMAX to be consistent with Fig. 5a (or the other way round?)

R: Yes. In the revised version we have included both 90 and 99th percentile for the German rain gauge stations (Figure 6). It should be noted however, that the observations for the Netherlands (Figure 5) are a gridded radar product, while those for Germany (Figure 6) are based on (non-gridded) station data. For this reason, the two plots will never be completely comparable.

- Fig. 7: again for the intensity metric, I would not use ALADIN in these plots, as it is unfair (see first comment). It would be so much easier for the reader to have the same metrics for Spain as for Germany and the Netherlands (also it takes a bit of time to understand this new metric!). Since you have hourly gauges available there, I guess it would be possible? Would you also be able to find hourly gauges in Norway to also do the same plot?

R: The comment about unfair comparison has been discussed above. We have also

discussed our use of the different metrics: it was our choice to do so in order to diversify the analyses and show different aspects of the model performance. We do not aim to compare the performance over the different domains because the modelling setup was quite different for different regions. For example, the Iberian Peninsula CPRCM was forced with ERA-Interim directly, while an intermediate model is used for all other domains. However, the intermediate models are also different for different regions, and the impact of the intermediate model can be substantial.

- For Spain, why are you not using SON, which is much more convective than JJA? (e.g. like Fumiere et al. (2019). A comparison with their results for southern France could be interesting: they found that AROME has a very good distribution but underestimates the strongest and rarest rain rates.)

R: In JJA precipitation is only convective but in SON it is mixed with large scale precipitation. Also, we wanted to show the precipitation specifically in summer because of its importance in the dry period, rather than the other wetter seasons. Finally, as stated above, the extent of analysis per domain is limited by the large number of domains that are shown, so certain choices had to be made. The Referee's suggestions will be addressed in a separate paper for this domain.

- In general, how are the simulations performing for lower precipitation values? (e.g. Berthou et al. (2018) found an underestimation of low-value precipitation in UKMO and to a lesser extend ETH-COSMO. These values are important to get a good climatology and often underestimated by convection-permitting models, which can also lead to soil moisture depletion in summer. It seems from Fig. 5c that there is a dry bias in the model?

R: The response to this comment is following our arguments from the first response above. Analysis of weak precipitation would have to be done for all the relevant domains, which would considerably increase the number of figures and amount of text. Furthermore, it would most likely invoke additional analyses to help explain the biases.

We could include this in the supplementary material if absolutely necessary, but we prefer to leave these analyses for the separate papers mentioned above.

Minor comments:

- P6 – lines 1-2: "this helps in decreasing. . .": the atmospheric models at 12 and 2.5km have very different rain rates (as you show later), so I doubt that the soil spin-up inherited from the 12km model actually helps: would you be able to show that your soil moisture is not drifting in the 2.5km simulations? You don't actually mention how long a spin-up you use for the 2.5km models.

R: Good point. This has been rewritten to: "This could help in decreasing the soil spin-up time, provided the precipitation climatology is similar for the two physics packages." We typically use 1 year of spin-up. We have discovered a bug in our soil interpolation code for the simulations presented here so we cannot show the correct time evolution of soil moisture, but since 1 year is used for the spin-up, this bug does not affect the final results.

- It is worth mentioning that results from Fig. 6 (overestimation of intense precip- itation (10-40mm/h) are consistent with the analysis of the UKMO/ETH-COSMO models in Berthou et al. (2018) for Germany.

R: We thank the reviewer for pointing this out. We have made a remark on this in the revised version.

- P10Lines15-17: which confidence interval are you using from the bootstrap re- sampling? Can you expand the method a bit?

R: We have now included an additional explanation of the method: "Confidence inter- vals (95%) are estimated using bootstrapping. The bootstrapping technique consists of constructing a large number of "synthetic" time-series by re-sampling days from the original time-series (sampling with replacement), and computing the target statistic for each of the resamples. This gives an empirical distribution for the target statistic, from

which the confidence interval can be estimated."

- P12 Lines17-20: "This indicates that observations. . . -> This indicates that the ALADIN biases in the distribution are probably larger than inter-decadal variability in the distribution.

R: We agree that this is one implication. However, since HCLIM38-AROME agrees quite well with the observations, the other implication is that the inter-decadal variability is probably sufficiently small to allow for the evaluation of HCLIM38-AROME (unless the agreement is by chance, where the potential bias in HCLIM38-AROME is of a similar sign and magnitude as the potential change in the inter-decadal variability). We have accordingly rephrased this sentence, and also gave it a somewhat milder ("seems" instead of "indicates") meaning: "Therefore it seems that the decadal variability in the distribution is sufficiently small to allow the comparison between observations and simulations of different (but close) periods in this case."

- P13, Lines 9-11: this is valid for all the simulations, move this to the start of section 3 (or remove it).

R: This has been removed.

- P13, Lines 32-33: does this mean that the biases are in a temperature range where actually it does not impact melting, or that diurnal cycle changes balance mean monthly changes? Can you expand a bit on this?

R: We have rewritten the sentence to clarify this: "As evident from Fig. 8b, despite the overall cold biases all three model versions achieve the correct number of days with melt at the three most southerly locations (both at the upper and lower stations). This indicates that even where the model monthly and seasonal mean temperatures are too low, they are still high enough to achieve the correct amount of days with melt."

REFERENCES: Crespi, A., Lussana, C., Brunetti, M., Dobler, A., Maugeri, M., and Tveito, O. E.: High-resolution monthly precipitation climatologies over Norway

(1981–2010): Joining numerical model data sets and in situ observations, International Journal of Climatology, 39, 2057–2070, https://doi.org/10.1002/joc.5933, 2019. Toivonen, E., Hippi, M., Korhonen, H., Laaksonen, A., Kangas, M., and Pietikäinen, J.-P.: The road weather model RoadSurf (v6.60b) driven by the regional climate model HCLIM38: evaluation over Finland, Geosci. Model Dev., 12, 3481–3501, https://doi.org/10.5194/gmd-12-3481-2019, 2019. Wu, M., Nikulin, G., Kjellström, E., Belušić, D., Jones, C., and Lindstedt, D.: The impact of RCM formulation and resolution on simulated precipitation in Africa, Earth Syst. Dynam. Discuss., https://doi.org/10.5194/esd-2019-55, in review, 2019.

Response to RC2

General Comments: The goal of this study is to introduce the HCLIM38 model, explain its origins, and describe the results from a number of simulations at different resolutions, using different physics packages. The authors provide ample evidence of performance over differing parts of the globe and over varying topography/climates. Overall, the simulation results show that HCLIM38 performs as well as or better (specifically at higher resolutions)than other RCM/CPRCMs. While the scientific evidence is strong, work is needed to clarify the origins of HCLIM38 and its different configurations to reduce confusion. In addition, I would streamline the discussion of small nuances of HCLIM38 as compared to many other related, but different NWP and GCM/RCM systems, since the latter aren't the focus of this paper. Focusing on the three configurations of HCLIM38 and the over-arching differences will improve clarity. With sufficient improvement to the manuscript, publication in Geoscientific Model Development should be considered.

Major Comments:

1. Section 2.1 (and beyond) is very confusing. Even after reviewing Table 1, I'm unable to make the right connections for each system. If HARMONIE is not an actual model,is it just HIRLAM-ALADIN with a scripting system? It would be good to specify

that HIRLAM-ALADIN is being phased out for AROME in the text (not in the table). Also, distinction between a model and its self-titled physics suite is necessary (AROME is a model, but has a physics suite with the same name). Line 3 states that HCLIM is based on ALADIN-HIRLAM, but line 8 says HARMONIE-AROME is the basis for HCLIM. A clear delineation between what is a true model, what is just a configuration of a model, what is a physics package, and how HCLIM38 fits into this picture is needed. I understand that a slew of changing acronyms is part of our work, but clarification of this section is necessary, and would go a long way toward reader comprehension.

R: We thank the Referee for many useful comments that helped to improve the clarity of the paper. We agree that the terminology in Section 2.1 was complex and difficult to follow. We have now rewritten Section 2.1 for better clarity, taking into account Referee's comments. The only exception is that we chose not to mention phasing out of the HIRLAM model in the text, since this is not relevant information for HCLIM. It is still mentioned in Table 1 for completeness, but now with better explanation. Namely, the HIRLAM model (note that HIRLAM can denote either the model or the community) has been replaced with the ALADIN-HIRLAM NWP model, in particular with the HARMONIE-AROME configuration.

2. It would be beneficial to have a table/plot showing how the three HCLIM38 configuration climate simulations were run. What did the domains look like (maybe providea plot for each)? What were the time steps and required wall clock times to run the simulations? What were the LBCs used and from what models (some of this is already in the text)? How often were the LBCs updated?

R: These data are now given in the updated Table S1 in Supplementary Material. The required wall clock time depends a lot on the modelling setup and the HPC that the model is running on, so we provide only rough estimates in Table S1. We do not provide the figures for all domains as this would take too much space, but we specify their approximate extents in Table S1.

[Figure]

Minor Comments:

1. Please expand upon/describe what "added value" is referring to in the first paragraph of the introduction. I assume this is referring to climate forecast accuracy, but it would be good to have a better idea of what metrics the authors have in mind.

R: The shortest description is that there are benefits in higher-order statistics. This includes all physical processes and surface characteristics that are not resolved by GCMs. The potential metrics are many, such as differences in power spectra, precipitation distributions, etc. The sentence now reads: "The added value, expressed as e.g. higher-order statistics, comes from the improved resolution of both regional physiography and atmospheric processes."

2. Please briefly summarize the overarching/major changes/improvements compared to older versions of HCLIM in the introduction.

R: This has been explained in a separate section: 2.4 Differences from HCLIM36.

3. Is there a hydrostatic vs. non-hydrostatic namelist option in HCLIM that is invoked when using the different physics packages (AROME vs. ALADIN or ALARO)?

R: Yes, it is a namelist switch.

4. More information is necessary on what the term "cycle" represents. What systems correspond to the "NWP model configurations" that are referred to as cycles? Which MET services are running them in real-time?

R: We now explain the origin and usage of the term "cycle" at the end of Section 2.1.

5. The authors discuss different model resolution configurations in 2.2, but then switch to physics packages with the same names as the models halfway through the paragraph. The delineation between the model and associated physics should be clear. Is the "HARMONIE-AROME" model the "similar NWP system" referenced in line 26 on Page 3? If so, it should be clarified and tied into the previous sentence.

[Figure]

R: We agree that mixing the terminology "model configuration" and "physics package" was confusing. This has been rewritten, and we now use the term "model configuration" throughout. When mentioned, a physics package is now treated as a part of a model configuration. We now state "the HARMONIE-AROME NWP system" instead of "similar NWP system".

6. The authors state that there is no deep convection parameterization, so it can only be used at convection-permitting scales. Are there any scale-aware schemes that could handle unresolved convection below 4 km?

R: We assume that the Referee refers to the description of AROME in 2.2.4. In the case of AROME, there is no convection parametrization except for the shallow convection. On the other hand, ALARO has the 3MT convection scheme that is scale-aware (described in 2.2.3).

7. Please describe how Fig. 2 was created. What is EOBS? Are these mean temperature/precip differences averaged across the whole 10-year period?

R: The reference to E-OBS gridded observations is given in the sentence before Fig. 2 is referenced. Yes, the plot shows differences in 10-year seasonal means of daily precipitation and 2 m temperature. The description of these fields is given in Fig. 2 caption.

8. It appears as though Crespi et al., 2019 used the in-situ observations to correct the bias of HCLIM38-AROME precipitation data over Norway in order to arrive at a more accurate precipitation climatology. It might be good to explain this detail.

R: We now include the following explanations in the same paragraph: "The high-resolution model data is used to provide a spatial reference field for an improved interpolation of the in situ observations, where there are no measurements. The result is neither a purely bias-corrected RCM data set, nor are the in situ observations corrected."

9. There is no Fig S1 or Table S1, but they are referred to in the text.

R: Fig. S1 and Table S1 are located in the Supplementary Material (hence the S). We will consult the GMD Editorial Office whether this needs to be specified more clearly.

10. Please briefly describe the exceedance plots and how you calculated them.

R: The exceedance plots are computed by simple ordering the FLDMAX or FLDMEAN data and inferring the empirical probability of a given value based on its order-position. The confidence interval is determined by computing a large number of bootstrap re-samples of the same length as the original data set (sampling days with replacement), which is used to construct a distribution of possible statistics. The 95% range of this distribution is used as confidence intervals. This is now explained in Section 3.1.3 Summer precipitation over the Netherlands and Germany.

11. Any idea why HCLIM36-ALADIN would have early biases for both the time of maximum precipitation and maximum hourly intensity (Fig. 7)? Something in the convective parameterization scheme?

R: This is a common effect happening in models that use convective parameterizations. We have now included references to studies addressing this issue in more depth (the references are given in Conclusion and outlook because a similar larger scale model behaviour is reported in other two subsections: 3.1.3 Summer precipitation over the Netherlands and Germany, and 3.3.1 HCLIM38-ALADIN performance over Africa): "HCLIM38-AROME is able to realistically simulate both the diurnal cycle and maximum intensity of sub-daily precipitation, which coarser RCMs or GCMs generally cannot accomplish because of the limitations of convection parameterizations (e.g., Brockhaus et al., 2008; Fosser et al., 2015; Prein et al., 2015)."

12. How did the CAMS based aerosol climatology improve HCLIM38 if it wasn't coupled with radiation or microphysics parameterizations? If they weren't coupled, weren't they just passive tracers?
R: The direct effect was included and this led to slight improvements in the overall performance of HCLIM38. In Rontu et al. (2019) the CAMS aerosol information was introduced to the model by replacing the existing climatology with CAMS. On one hand this was only an initial step in a longer development chain and thus, the authors applied the existing structure of the older climatology (e.g. lumped some CAMS species together), but on the other hand this approach enabled the CAMS direct effect in the simulations. The work started in Rontu et al. (2019) continues in terms of introducing the aerosol inherent optical properties, using every CAMS species separately (no lumping together) and using the CAMS aerosol information in the cloud microphysics part. The Referee makes a good point that the text is misleading. We have improved the text and made it more clear.

Introduction - Are there scale-aware physics parameterizations in GCMs, RCMs, and CPRCMS (as there are in NWP) that could handle increasing resolutions of climate modelling and could shut themselves off appropriately?

R: Yes and one such parametrization is present in HCLIM38, as explained in section 2.2.3 ALARO. We now mention this option in Introduction: "An alternative approach is to use scale-aware convection parametrizations that can adjust their effects based on the fraction of resolved convection in a model grid box (e.g., Gerard et al., 2009)."

Introduction - Are the terms convection-"resolving" and convection-"permitting" interchangeable for CPRCMS?

R: Some researchers criticize the term "convection resolving" because CPRCMs do not resolve the entire spectrum of moist convective features. The term "convection permitting" therefore provides a more accurate description of the model's capability. For consistency, we have decided to use only one term in the paper, and it is "convection permitting".

Page 2, Line 18 – I might expand briefly upon why nesting from O(10km) to O(1km) is not possible, aside from the reference, since this is regularly done in NWP.

R: As the reviewer states, O(10km) to O(1km) is not a problem, but L18 states that going from GCM to CPRCM is a too large step without nesting. This corresponds to O(100km) to O(1km).

Page 3, Line 21 – Climate or NWP limited-area models?

R: This statement refers to both climate and NWP models, hence it is given in general form.

Page 3, Line 26-29 – ALADIN or ALARO physics, model, or both? This needs to be clear.

R: "Physics packages" has been replaced with "model configurations" for clarity, in line with Referee's comment 5 and other such changes in Section 2.1 and 2.2.

Page 6, Line 2 – I assume it be used with GCMs and not just RCMs for LBCs?

R: We have now replaced "different RCMs" with "other climate models" in the referred sentence.

Page 8, Line 3-4 – EOBS and PRUDENCE are not defined.

R: EOBS (now corrected to E-OBS) is used as such without a definition even in the original papers and the web site, so we follow their strategy and provide only the reference. PRUDENCE is now defined.

Page 6, Line 23 – I'm not sure what the "e.g." is doing here.

R: Only certain components of the earth system are mentioned there, so with "e.g." we wanted to emphasize that there can be other components that are coupled to the climate model. This has been rewritten to "...when CNRM-ALADIN is coupled with other components of the Earth system such as aerosols (Drugé et al., 2019), oceans and rivers..."

Page 10, Line 31 – There is no Fig. S3 as referred to in the text.

R: See previous comments.

Page 13, Line16 – There is no Fig. S5 as referred to in the text.

R: See previous comments.

Response to SC1

A short comment to update you with the latest references concerning the use of AL-ADIN and AROME in climate mode at CNRM. So probably some of those references can fit in your section 2.3. I let you decide what is relevant. - I think that it is worse mentioning that ALADIN has been used in climate mode for more than 10 years (see for example, Radu et al. 2008, Déqué and Somot 2008, Colin et al. 2010) in particular in projects such as ENSEMBLES or CORDEX. - Before being used in cycle 41t1 (Coppola et al. 2019), AROME has been used in cli- mate mode at CNRM with the cycle 38 (same cycle you are describing in this GMDD) to explore the added-value of such CPRCM with respect to a twin 12km ALADIN simulation. References are: Déqué et al. 2016, Fumière et al. 2019 - There are references for the use of ALADIN in a fully coupled framework, the so- called RCSM configuration including the sea and river representation. For example, Sevault et al. 2014 (CNRM-RCSM4) and Darmaraki et al. 2019 (CNRM-RCSM6 the latest version based on ALADIN v6.3).

R: We are grateful for notifying us about these updates. They have been included in the manuscript.

---

## Author Response (AR2)

Response to the Editor

Please find below our responses to Editor's comments (marked with **R**), and the marked-up manuscript version.

*As to the present manuscript under consideration... the authors made the code accessible to me. I cannot identify the exact set up files for the experiments conducted in this paper, and I would like to see them clearly identified and made available as a supplement to the paper.*

**R:** We have included the domain definition file and an example of experiment configuration file in Supplementary material. The configuration file can be modified using the information listed in Table S1. This is now discussed in Sec. 2.1:

"Configuration information for the experiments analysed here can be found in Supplementary material. The domain definition file (Harmonie_domains.pm) together with a configuration file (config_exp.h) include all information about an experiment setup. The options used to modify the configuration file for the different experiments are listed in Table S1."

*I am still a bit confused (maybe more so after admiring the code!) on the naming of the different model setups. Why is it "HARMONIE-AROME" but not "HARMONIE-ALARO", and then we have HCLIM38-AROME, HCLIM38-ALADIN, and HCLIM38-ALARO... but, HARMONIE-AROME=HCLIM38-AROME ?*

**R:** The terminology is indeed quite complex and not always easy to describe. There is only HARMONIE-AROME because the HIRLAM consortium is officially maintaining and developing only the AROME model configuration for NWP purposes. Nevertheless, other model configurations are available in the code and can be used, even though they are not officially supported. This is why HCLIM38 can include all the three model configurations. HARMONIE-AROME refers to a specific model configuration for NWP purposes. HCLIM38-AROME is similar to (and based on) HARMONIE-AROME but includes modifications that are necessary for climate simulations, such as a more sophisticated soil scheme.

*An email exchange revealed an amazing website full of helpful information for users of the model which includes archived information for recent model versions. I would like to see this website discussed in the manuscript.*

**R:** The website is now mentioned in Sec. 2.1. Please see the next response too.

*I would also like to know how the version of the model (along with the documentation) discussed in this paper are to be archived such that future readers of the paper will be able to obtain the precise version. This should be clarified in the manuscript.*

**R:** We have included the information about code and documentation archiving in Sec. 2.1:

[revised manuscript text omitted]

**Supplementary material**

[Figure]

**Figure S1.** Winter (DJF) and summer (JJA) (a) near-surface temperature and (b) precipitation anomalies, comparing HCLIM cycle 36 and 38 with E-OBS for 1999–2007. E-OBS is available only over land.

[Figure]

**Figure S2.** HCLIM36 and HCLIM38 monthly anomalies compared to ERA-Interim of (a), (c) surface short-wave (SWd) and long-wave (LWd) down-welling radiation, and (b), (d) latent (LH) and sensible (SH) heat fluxes, for 1999–2007. Shown are averages over two regions: (a), (b) Scandinavia and (c), (d) South-East Europe.

**Figure S3.** As in Fig. 5 (in the main text), but for upscaled precipitation statistics. Prior to analysis the input-data has been re-gridded to same resolution as the the driving RCM (12 km). The labels in the figure now carry an additional "Up11" in their name to indicate that upscaling has been done prior to analysis.

[Figure]

**Figure S4.** Available German rain gauges (circles) and those used in the analysis (filled circles, altitudes below 500 m) with coloring indicating altitude.

[Figure]

**Figure S5.** Observations used in the precipitation evaluation over Spain (black points).

[Figure]

**Figure S6.** (a) Annual mean of near-surface temperature (°C) from CRU observations and (b) the corresponding differences between HCLIM38-ALADIN and CRU. (c) Annual mean of daily precipitation (mm day$^{-1}$) from GPCC observations and (d) the corresponding differences between HCLIM38-ALADIN and GPCC. The analysed period is 1980–2010 and HCLIM38-ALADIN grid spacing is 50 km.

**Table S1.** The summary of HCLIM38 experiments in this study.

| Experiment | Period | Simulation domain[1] | Resolution in km (time step in s) | Model[2] | Lateral boundary data[3] (update interval in h) |
|---|---|---|---|---|---|
| Pan-Europe | 1999-2008 | Reduced Eur N (REU12_N) | 12 (300) | HCLIM38-ALADIN | ERA-Interim (6) |
| Norway | 2004-2016 | NORWAY2.5 | 2.5 (60) | HCLIM38-AROME | HCLIM38-ALARO (6) |
| | | Large pan-Alpine (PALP2.5) | 2.5 (60) | HCLIM38-AROME | RACMO2 (1) |
| The Netherlands and Germany | 2000-2009 | Reduced Eur S (REU12_S) | 12 (300) | HCLIM38-ALADIN | ERA-Interim (6) |
| | | Pan-Alpine (PALP3) | 3 (75) | HCLIM38-AROME | HCLIM38-ALADIN (3) |
| Iberian Peninsula | 2005-2014 | REU12_S | 12 (300) | HCLIM38-ALADIN | ERA-Interim (6) |
| | 1990-1999 | IBERIA2.5 | 2.5 (60) | HCLIM38-AROME | ERA-Interim (6) |
| | | | 24 (600) | HCLIM38-ALARO | ERA-Interim (6) |
| Arctic region | Summer of 2014 | ARCTIC24/12 | 24 (600) | HCLIM38-ALADIN | ERA-Interim (6) |
| | | | 12 (300) | HCLIM38-ALADIN | ERA-Interim (6) |
| Africa | 2000-2009 | AFRICA50 | 50 (1200) | HCLIM38-ALADIN | ERA-Interim (6) |
| The Lake Victoria Basin | 2005-2006 | EAFR25/12.5 | 25 (600) | HCLIM38-ALADIN | ERA-Interim (6) |
| | | | 12.5 (360) | HCLIM38-ALADIN | ERA-Interim (6) |
| | | LVIC2.5 | 2.5 (60) | HCLIM38-AROME | HCLIM38-ALADIN (3) |

[1] Domains (with the names in capital letters) are described in the attached file Harmonie_domains.pm

[2] Core hours per simulated month per grid point multiplied with the time step are about 0.75 (1.1) for ALADIN (AROME)

[3] The regional models providing lateral boundary data have used ERA-Interim at their lateral boundaries

---

## Author Response (AR3)

Response to the Editor

Please find below our response to Editor's comment (marked with **R**), and the marked-up manuscript version.

*Comments to the Author:*
*I am sorry to make another minor revision but experience shows that if I choose technical corrections, the requested changes are rarely attended to. Please feel free to email me to prompt me into action if you don't hear back swiftly after your next response!*

*You wrote in the authors response,*
*" R: The terminology is indeed quite complex and not always easy to describe. There is only HARMONIE-AROME because the HIRLAM consortium is officially maintaining and developing only the AROME model configuration for NWP purposes. Nevertheless, other model configurations are available in the code and can be used, even though they are not officially supported. This is why HCLIM38 can include all the three model configurations. HARMONIE-AROME refers to a specific model configuration for NWP purposes. HCLIM38-AROME is similar to (and based on) HARMONIE-AROME but includes modifications that are necessary for climate simulations, such as a more sophisticated soil scheme."*

*This is so much more comprehensible than what I find in the manuscript. Such practical details are absolutely appropriate to include in a GMD paper. So please can you add the information to the manuscript…*

**R:** We have now included the requested modifications in the manuscript.

[revised manuscript text omitted]
 is a scripting system developed and used by HIRLAM countries for operational NWP applications. All the above-mentioned ALADIN-HIRLAM model configurations are available in the HARMONIE scripting system, but only the specific configuration of AROME (HARMONIE-AROME) is officially supported, developed and used in HIRLAM NWP applications (Bengtsson et al., 2017).

The HCLIM climate model development is based on the HARMONIE system. The HARMONIE-AROME model configuration is designed for convection permitting scales and is used with nonhydrostatic dynamics, which is the primary focus of HCLIM development. The model configurations ALADIN and ALARO are also used in HCLIM applications, typically for coarser resolutions with hydrostatic dynamics. Since HCLIM includes these three different model 
[revised manuscript text omitted]